# Decentralized and Lifelong-Adaptive Multi-Agent Collaborative Learning

## Abstract

Decentralized and lifelong-adaptive multi-agent collaborative learning aims to enhance collaboration among multiple agents without a central server, with each agent solving varied tasks over time. To achieve efficient collaboration, agents should: i) autonomously identify beneficial collaborative relationships in a decentralized manner; and ii) adapt to dynamically changing task observations. In this paper, we propose `DeLAMA`, a decentralized multi-agent lifelong collaborative learning algorithm with dynamic collaboration graphs. To promote autonomous collaboration relationship learning, we propose a decentralized graph structure learning algorithm, eliminating the need for external priors. To facilitate adaptation to dynamic tasks, we design a memory unit to capture the agents' accumulated learning history and knowledge, while preserving finite storage consumption. To further augment the system's expressive capabilities and computational efficiency, we apply algorithm unrolling, leveraging the advantages of both mathematical optimization and neural networks. This allows the agents to 'learn to collaborate' through the supervision of training tasks. Our theoretical analysis verifies that inter-agent collaboration is communication efficient under a small number of communication rounds. The experimental results verify its ability to facilitate the discovery of collaboration strategies and adaptation to dynamic learning scenarios, achieving a 98.80% reduction in MSE and a 188.87% improvement in classification accuracy. We expect our work can serve as a foundational technique to facilitate future works towards an intelligent, decentralized, and dynamic multi-agent system.

## 1 Introduction

Collaboration is a stealthy, yet ubiquitous phenomenon in nature, evident in the cooperative behaviors of animals and humans, from pack-hunting to constructing complex social relationships. Such cooperative behaviors enable individuals to accomplish complex tasks (Woolley et al., 2010) and form social relationships (Mennis, 2006). Inspired by this natural tendency for collaboration, the field of multi-agent collaborative learning has been extensively explored (Mennis, 2006). It aims to enable multiple agents to collaboratively strategize and achieve shared objectives, exhibiting capabilities surpassing that of any single agent. Recently, there has been an urgent demand for relevant methods and systems, such as vehicle-to-everything(V2X) communication-aided autonomous driving (Jung et al., 2020; Li et al., 2022), multi-robot environment exploration (Burgard et al., 2005; 2000) and collaborative training of machine learning models across multiple clients (McMahan et al., 2017).

To promote efficient collaboration among the agents, it is crucial to identify whom to collaborate with. This could avoid redundant and low-quality information from an arbitrary collaborator, improving collaboration effectiveness. Current collaboration methods often involve using predefined fixed collaboration relationships, such as fully connected graphs (e.g., CoLLA (Rostami et al., 2017), DiNNO (Yu et al., 2021)), star graphs with a central server (e.g., federated learning (McMahan et al., 2017)), or task correlations calculated by a central server in multi-task learning (Liu et al., 2017). However, these predefined structures are limited by two major issues. First, collaboration (McMahan et al., 2017; Liu et al., 2017) guided by a central server faces vulnerability issues, where the central server failure would disable all collaborations. Second, the static collaboration (McMahan et al., 2017; Liu et al., 2017) relationships limit the flexibility and efficiency of the system performance when faced with dynamic scenarios. In dynamic scenarios (Wang et al., 2023), agents come up with continuous and time-varying observations and update their local decision-making model, leading to

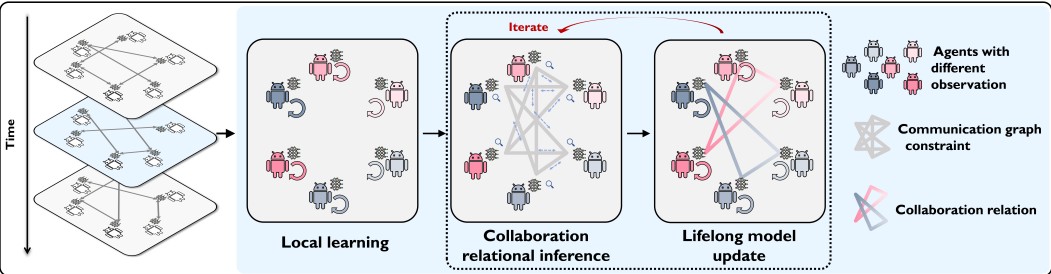

Figure 1: The decentralized and lifelong-adaptive multi-agent collaborative learning system. The system consists of three steps: local learning, collaborative relational inference, and lifelong model update. In the first step, each agent learns model parameters from its own observations. In the second step, agents share their parameters via a communication graph to learn collaboration relationships. In the final step, agents exchange parameters with collaborators and refine their models. Steps two and three repeat until convergence.

the change of agent potential relationships. The static collaboration modeling becomes inadequate and inflexible to capture the relationships in these dynamic scenarios, limiting the collaboration efficiency. In this work, we introduce a novel decentralized and life-long adaptive learning framework for multi-agent cognitive-level collaboration. The core feature of our framework is that each agent can autonomously choose its collaborators and evolve over time.

To address the issue of the static collaboration relationship and centralized collaboration, we design a lifelong-learning-based approach for multi-agent decentralized collaboration. The key approach is enabling agents to dynamically adapt to ever-changing observations during individual model training, providing a basis for obtaining more accurate collaboration relationships. In agent individual model training, we design a memory unit to capture an agent's accumulated learning history and knowledge, facilitating adaptation to evolving observation flows. Compared with the static collaboration mechanism (McMahan et al., 2017; Liu et al., 2017), this lifelong-learning-based approach could lead to evolving collaboration structures and model parameters, making it more suitable for dynamic scenarios.

Our core objective is to provide a theoretically interpretable collaborative learning algorithm rather than end-to-end training. To achieve this, we propose `DeLAMA`, a decentralized and lifelong adaptive multi-agent learning algorithm derived from mathematical optimization. To further promote more learning ability and flexibility, we apply algorithm unrolling techniques (Monga et al., 2021; Kamilov et al., 2023) to upgrade the aforementioned iterative algorithm of the decentralized optimization to a neural network. Through algorithm unrolling, each iteration of our optimization solution is mapped to a customized neural network layer with learnable parameters. The network parameters are learned via supervised training tasks, enabling agents to effectively learn to collaborate. This approach combines the advantages of both mathematical optimization and neural networks; see the illustration in **Figure** 1.

## 2 OPTIMIZATION FOR DECENTRALIZED AND LIFELONG-ADAPTIVE COLLABORATIVE LEARNING

We consider a setting with $N$ collaboration agents (McMahan et al., 2017), each agent can train a dynamic machine-learning model for evolving tasks. At any given timestamp $t$, the $i$-th agent receives a training dataset $\mathcal{D}_i^{(k)}$. The agents could communicate with each other according to the collaboration relationship $\mathbf{W}^{(t)}$. Here we use $\mathbf{C}^{(t)}$ to represent communication constraint between the agents; see Appendix A for details. In practice, accessing the collaboration relationships $\mathbf{W}^{(t)}$ among agents is challenging due to i) the difficulties of quantifying task similarity; ii) the need to adapt to dynamically evolving tasks among agents; and iii) the necessity for a decentralized and autonomous mechanism to determine this structure. Hence, we incorporate the learning of collaboration graph into the optimization problem (1) where agents actively choose their collaborators, which takes both

$\mathbf{W}^{(t)}$ and $\boldsymbol{\Theta}^{(t)}$ into consideration:

$$\min_{\boldsymbol{\Theta}^{(t)},\mathbf{W}^{(t)}} \sum_{i=1}^{N} \mathcal{L}_i^{(t)}\left(\boldsymbol{\theta}_i^{(t)}\right) + \lambda_1 \left\|\boldsymbol{\theta}_i^{(t)}\right\|_2^2 + \mathcal{I}_{\geq 0}\left(\mathbf{W}^{(t)}\right) + \lambda_2 \,\mathbf{tr}\left(\boldsymbol{\Theta}^{(t)\top}\mathbf{L}^{(t)}\boldsymbol{\Theta}^{(t)}\right) + \lambda_3 \left\|\mathbf{W}^{(t)}\right\|_F^2$$

$$\text{s.t. } \left\|\mathbf{W}^{(t)}\mathbf{1}\right\|_1 = \boldsymbol{m},\; \mathbf{diag}\left(\mathbf{W}^{(t)}\right) = \mathbf{0},\, \mathbf{W}_{ij}^{(t)} = 0 \text{ if } \mathbf{C}_{ij}^{(t)} = 0,\; 1 \leq i,j \leq N. \tag{1}$$

Here the accumulated loss of the $i$-th agent with the model parameter $\boldsymbol{\theta}_i^{(t)}$ is $\mathcal{L}_i^{(t)}\left(\boldsymbol{\theta}_i^{(t)}\right) = \frac{1}{t}\sum_{k=1}^{t} \ell^k\left(\boldsymbol{\theta}_i^{(t)}\right)$, where $\ell^k\left(\boldsymbol{\theta}_i^{(t)}\right) = -\log p\left(\mathcal{D}_i^{(k)}|\boldsymbol{\theta}_i^{(t)}\right)$ is the loss of the model $\boldsymbol{\theta}_i^{(t)}$ evaluated on the supervised dataset $\mathcal{D}_i^{(k)}$. The edge weights of the collaboration graph $\mathbf{W}^{(t)}$ are set to be positive without self-loops by adding the indicator function $\mathcal{I}_{\geq 0}\left(\mathbf{W}^{(t)}\right)$ and the constraint $\mathbf{diag}\left(\mathbf{W}^{(t)}\right) = \mathbf{0}$ to the optimization.

Note that: i) to increase the generalization ability for models' parameter learning, we incorporate the $L_2$ regularization term associated with parameter $\lambda_1$; ii) to ensure non-trivial solutions and smooth edge weights in graph structure $\mathbf{W}^{(t)}$, we introduce the $L_2$ regularization term of $\mathbf{W}^{(t)}$ to the optimization problem with hyper-parameter $\lambda_3$; and iii) to maintain a constant norm and sparse structure in the collaboration graph, we normalize the graph $\mathbf{W}^{(t)}$ to hyper-parameter $\boldsymbol{m}$ using $L_1$ metric.

## 2.1 ANALYTICAL SOLUTION FOR THE OPTIMIZATION PROBLEM

**Understanding the solution process.** The solution process of **Problem** (1) can be described as a recursive function $\mathcal{F}(\cdot)$ that takes both the training data $\mathcal{D}_{1:N}^{(t)}$ and the learning memories $\mathcal{M}_{1:N}^{(t-1)}$ as input then outputs the model parameters $\boldsymbol{\Theta}^{(t)}$ and collaboration relationships $\mathbf{W}^{(t)}$,

$$\boldsymbol{\Theta}^{(t)}, \mathbf{W}^{(t)}, \mathcal{M}_{1:N}^{(t)} = \mathcal{F}\left(\mathcal{D}_{1:N}^{(t)}, \mathcal{M}_{1:N}^{(t-1)} \big| \mathbf{C}^{(t)}\right), \tag{2}$$

where $\mathbf{C}^{(t)}$ is the communication structure constraint at time $t$. Specifically, due to the existence of the graph smoothness term $\mathbf{tr}\left(\boldsymbol{\Theta}^{(t)\top}\mathbf{L}^{(t)}\boldsymbol{\Theta}^{(t)}\right)$ we leverage alternate convex search between $\boldsymbol{\Theta}^{(t)}$ and $\mathbf{W}^{(t)}$ since the original problem is non-convex. Based on this approach, our solver $\mathcal{F}(\cdot)$ can be decomposed into three parts: i) **local learning**, this step initializes the model parameters $\boldsymbol{\Theta}^{(t)}$ and updates the learning history $\mathcal{M}_{1:N}^{(t)}$; ii) **collaborative relational inference**, this step learns the collaboration structure $\mathbf{W}^{(t)}$ given fixed $\boldsymbol{\Theta}^{(t)}$; and iii) **lifelong model update**, this step optimizes the model parameters $\boldsymbol{\Theta}^{(t)}$ under the collaboration structure $\mathbf{W}^{(t)}$.

**Local learning.** In this part, the algorithm tries to give an initialization $\boldsymbol{\Theta}_{init}^{(t)}$ of $\boldsymbol{\Theta}^{(t)}$. Here we use Taylor expansion by solving the following first order condition:

$$\frac{\partial \mathcal{L}_i^{(t)}\left(\boldsymbol{\theta}_i^{(t)}\right)}{\partial \boldsymbol{\theta}_i^{(t)}} = \frac{1}{t}\sum_{k=1}^{t} \mathbf{H}_i^{(k)}\left(\boldsymbol{\theta}_i^{(t)} - \boldsymbol{\alpha}_i^{(k)}\right) + \boldsymbol{g}_i^{(k)} = \mathbf{A}^{(t)}\boldsymbol{\theta}_i^{(t)} - \boldsymbol{b}_i^{(t)} = \mathbf{0} \tag{3}$$

where $\mathbf{H}_i^{(k)}$ and $\boldsymbol{g}_i^{(k)}$ are second-order & first-order information of $\ell^k\left(\boldsymbol{\theta}_i^{(t)}\right)$ at Taylor expansion point $\boldsymbol{\alpha}_i^{(k)}$. Here $\mathcal{M}_i^{(t)} = \left(\mathbf{A}_i^{(t)}, \boldsymbol{b}_i^{(t)}\right)$ is the running average of historical $\mathbf{H}_i^{(k)}$s and $\boldsymbol{g}_i^{(k)}$s, serving as the learning history. Although this represents an approximate solution, we provide a theoretical proof demonstrating that the approximation error is minimized when the expansion point $\boldsymbol{\alpha}_i^{(t)}$ is set to $\mathbf{0}$, thereby offering a relatively effective and numerical stability .

**Theorem 1** *Let* $\mathbf{H}\left(\boldsymbol{\alpha}_i^{(t)}\right) = \nabla^2_{\boldsymbol{\alpha}_i^{(t)}} \mathcal{L}_i^{(t)}\left(\boldsymbol{\theta}_i^{(t)}\right)$ *be Lipschitz continuous with non-zero constant and* $\exists M$ *such that* $1 \leq k\left(\mathbf{H}\left(\boldsymbol{\alpha}_i^{(t)}\right)\right) \leq M$ *where* $k$ *is the conditional number of* $\mathbf{H}\left(\boldsymbol{\alpha}_i^{(t)}\right)$*, then*

$$\arg\min_{\boldsymbol{\alpha}_i^{(t)}} \mathbb{E}_{\boldsymbol{\theta}_i^{(t)*}}\left(\boldsymbol{tr}\left(\mathbf{H}\left(\boldsymbol{\alpha}_i^{(t)}\right)\right)^{-1} \left\|\boldsymbol{\alpha}_i^{(t)} - \boldsymbol{\theta}_i^{(t)*}\right\|_2^2\right) = \mathbf{0}.$$

Here the objective function serves as an approximation for the Riemannian distance between $\boldsymbol{\theta}_{i,init}^{(t)}$ and $\theta_i^{(t)*}$; see **Appendix** C for the optimizations, solutions and **Appendix** G.1 for detailed analysis and proof.

**Collaborative relational inference.** In this part, the algorithm tries to solve the sub-optimization problem of $\mathbf{W}^{(t)}$ given fixed $\boldsymbol{\Theta}^{(t)}$. It aims to find the target collaboration structure by solving the following optimization problem:

$$\min_{\mathbf{W}^{(t)}} \lambda_2 \, \mathbf{tr} \left( \boldsymbol{\Theta}^{(t)\top} \mathbf{L}^{(t)} \boldsymbol{\Theta}^{(t)} \right) + \mathcal{I}_{\geq 0} \left( \mathbf{W}^{(t)} \right) + \lambda_3 \left\| \mathbf{W}^{(t)} \right\|_{\mathrm{F}}^2 \tag{4}$$

$$\text{s.t. } \|\mathbf{W}^{(t)}\|_1 = \boldsymbol{m}, \ \mathbf{diag} \left( \mathbf{W}^{(t)} \right) = 0, \mathbf{W}_{ij}^{(t)} = 0 \text{ if } \mathbf{C}_{ij}^{(t)} = 0, \ 1 \leq i, j \leq N,$$

where $\mathcal{I}_{\geq 0} \left( \mathbf{W}^{(t)} \right)$ is the indicator function requiring edge weights of $\mathbf{W}^{(t)}$ to be positive. Note that different other graph structure learning approaches with centralized server, here the solution process should be fully decentralized. similar to the reformulation trick shown in (Pu et al., 2021), we define the block correspond to the $i$-th agent be $\boldsymbol{w}_i^{(t)} \in \mathbb{R}^{N-1}$ as the $i$-th row of the adjacency matrix $\mathbf{W}^{(t)}$ except the $i$-th element. With basic mathematical optimization, the solution is

$$\boldsymbol{w}_i^{(t)} = \mathbf{ReLU} \left( -\frac{\lambda_2 \boldsymbol{d}_i^{(t)} + z\mathbf{1}}{2\lambda_3} \right), \tag{5}$$

where $\boldsymbol{d}_i^{(t)}$ is defined as the distance between other models written by $\boldsymbol{d}_{ij}^{(t)} = \left\| \boldsymbol{\theta}_i^{(t)} - \boldsymbol{\theta}_j^{(t)} \right\|_2^2 / \mathbf{C}_{ij}^{(t)}$. See **Appendix** D for the consensus protocol of the agents and the optimization details.

**Lifelong model update.** Given the optimized collaboration relation $\mathbf{W}^{(t)}$, this part tries to find the model parameters $\boldsymbol{\Theta}^{(\mathbf{t})}$ by solving the corresponding non-constraint optimization problem:

$$\min_{\boldsymbol{\Theta}^{(t)}} \sum_{i=1}^{N} \mathcal{L}_i^{(t)} \left( \boldsymbol{\theta}_i^{(t)} \right) + \lambda_1 \left\| \boldsymbol{\theta}_i^{(t)} \right\|_2^2 + \lambda_2 \, \mathbf{tr} \left( \boldsymbol{\Theta}^{(t)\top} \mathbf{L}^{(t)} \boldsymbol{\Theta}^{(t)} \right). \tag{6}$$

Note that to ensure the decentralized collaboration mechanism, both iterative steps $\boldsymbol{\Phi}_{\mathrm{graph}}(\cdot)$ and $\boldsymbol{\Phi}_{\mathrm{param}}(\cdot)$ should be operated under the communication graph $\mathbf{C}^{(t)}$ constraint. Optimizing $\boldsymbol{\Theta}^{(t)}$ corresponds to solving a linear system with $N$ first-order conditions. To obtain the decentralized computation and reduce the computational overhead, we consider Jacobi-iteration Golub and Van Loan (2013) based approaches. Specifically, assuming the symmetry of the adjacency matrix $\mathbf{W}^{(t)}$, the solution is defined as

$$\left( \mathbf{A}_i^{(t)} + 2\lambda_1 \mathbf{I} + 4\lambda_2 \mathbf{D}_i^{(t)} \right) \boldsymbol{\theta}_i^{(t)} = \boldsymbol{b}_i^{(t)} + 4\lambda_2 \sum_{j=1}^{N} \mathbf{W}_{ij}^{(t)} \boldsymbol{\theta}_j^{(t)}, \tag{7}$$

where $\mathbf{D}_i^{(t)}$ is the degree of the $i$-th agent of the collaboration relation $\mathbf{W}^{(t)}$. We have the following theoretical guarantee about the proposed method.

**Theorem 2** *Let $\boldsymbol{\Theta}^{(t),k}$ be the model parameters in the $k$-th iteration of the linear equation, $\boldsymbol{\Theta}^{(t)*}$ is the target solution satisfying equation 7. For any $\epsilon$, if $\left\| \boldsymbol{\Theta}^{(t)} - \boldsymbol{\Theta}^{(t)*} \right\|_{\mathbf{F}}^2 \leq \epsilon$, then $k \sim \mathcal{O} \left[ \log \left( \frac{1}{\epsilon} \right) \right]$.*

**Theorem** 2 shows that model parameters can converge to the optimum with linear rate, demonstrating the computational efficiency of our approach; see **Appendix** G.2 for detailed proof.

## 3 Algorithm Unrolling for Decentralized and Lifelong-Adaptive Multi-Agent Learning

While the iterative algorithm enables agents to learn and collaborate, it encounters certain challenges that limit its real-world applicability. First, the Taylor expansion in $\boldsymbol{\Phi}_{\mathrm{local}}(\cdot)$ has potential approximation errors for non-linear functions, limiting the method's expressive capability for non-linear learning tasks. Second, the collaborative learning system in equation 1 has many hyper-parameters, demanding significant effort for searching and tuning hyper-parameters. Third, the algorithm requires numerous iterations. Single graph learning and message passing steps involve many iterations and alternating convex search further requires alternate iterations between these two steps. These numerous iterations cause significant communication overhead, making the algorithm less practical for real applications.

Figure 2: The unrolled network structure of the collaboration system with input data at one exact time. For each agent, the training data is firstly passed into a feed-forward network to be transformed into embeddings, and then this embedding is used to calculate the initialized model parameters according to $\Phi_{\text{local}}(\cdot)$. Then the agents start to communicate and collaborate to find proper parameters according to the iterations shown in $\Phi_{\text{graph}}(\cdot)$ and $\Phi_{\text{param}}(\cdot)$. Finally, the network $\mathcal{F}_\gamma(\cdot)$ outputs parameters $\Theta^{(t)}$. These output model parameters $\Theta^{(t)}$ are supervised by the data $(\mathcal{X}, \mathcal{Y}) \sim \mathcal{T}_i^{train}$. The training process of the network $\mathcal{F}_\gamma(\cdot)$ is learning to tune the parameter $\theta$ to find the optimal parameter learning strategies of $\Theta^{(t)}$.

In this section, we first describe the structure of the unrolled network, then introduce the learning-to-learn approaches to train the hyper-parameters of this network.

## 3.1 UNROLLING NETWORK DESIGN

Recall the compact solution shown in equation (2), which is a mechanism to learn model parameters given training data $\mathcal{D}_{1:N}^{(t)}$. We aim to use the algorithm unrolling techniques to transform equation 2 into the following neural-network-based unrolled mapping with learnable hyperparameters $\gamma$:

$$\Theta^{(t)}, \mathbf{W}^{(t)}, \mathcal{M}_{1:N}^{(t)} = \mathcal{F}_\gamma\left(\mathcal{D}_{1:N}^{(t)}, \mathcal{M}_{1:N}^{(t-1)}\big|\mathbf{C}^{(t)}\right). \tag{8}$$

The inputs are the training data and the output is the model parameters $\Theta^{(t)}$. This can be understood as using a neural network $\mathcal{F}_\gamma(\cdot)$ to learn the model parameter $\Theta^{(t)}$, where $\gamma$ is the hyperparameter controlling the network $\mathcal{F}_\gamma(\cdot)$ to obtain model parameters $\Theta^{(t)}$.

Specifically, we upgrade the iterative steps of equation (2) to the unrolled network layers associated with $\mathcal{F}_\gamma(\cdot)$:

$$\Theta^{(t),0}, \mathcal{M}_{1:N}^{(t)} = \Phi_{\text{local}}\left(\mathcal{D}_{1:N}^{(t)}, \mathcal{M}_{1:N}^{(t-1)}; \lambda_1, \beta\right), \tag{9a}$$

$$\mathbf{W}^{(t)} = \Phi_{\text{graph}}\left(\Theta^{(t),0}\big|\mathbf{C}^{(t)}; \lambda_2, \lambda_3\right), \tag{9b}$$

$$\Theta^{(t)} = \Phi_{\text{param}}\left(\Theta^{(t),0}, \mathbf{W}^{(t)}, \mathcal{M}_{1:N}^{(t)}\big|\mathbf{C}^{(t)}; \lambda_1, \lambda_2\right), \tag{9c}$$

where $\gamma = \left(\beta, \{\lambda_i\}_{i=1}^3\right)$ is the learnable parameters of network $\mathcal{F}_\gamma(\cdot)$. Here $\{\lambda_i\}_{i=1}^3$ are learnable hyperparameters and $\beta$ is the model parameter of a neural network. $\Phi_{\text{local}}, \Phi_{\text{graph}}, \Phi_{\text{param}}$ stand for local learning, collaborative relational inference and lifelong model update.

Note that: 1) to obtain a framework with fewer computations and less communication overhead, we reduce the number of iterations of alternative convex search between $\Phi_{\text{graph}}(\cdot)$ and $\Phi_{\text{param}}(\cdot)$ to 1 since the nonlinear operation corresponding to $\Phi_{\text{param}}(\cdot)$ is capable to find better initialization of model parameters $\Theta^{(t),0}$, which could simplify the process of finding collaboration relationships and model update, hence requires fewer iterations to converge; and 2) to autonomously tune the hyperparameters $\{\lambda_i\}_{i=1}^3$, we leverage the supervised training paradigm of algorithm unrolling by treating the hyperparameters as learnable parameters.

To increase the learning flexibility and expressive power of the framework, according to the calculate Hessian step of $\Phi_{\text{local}}(\cdot)$, we add an non-linear backbone network $\phi_\beta$ with learnable parameters $\beta$

Table 1: The system performance analysis on classification tasks of four different settings: Lifelong learning, federated learning, decentralized optimization, decentralized and lifelong-adaptive collaborative learning. `DeLAMA-WC` stands for `DeLAMA` without collaboration among agents and `DeLAMA-WM` stands for `DeLAMA` without lifelong-adaptive learning capabilities.

| Setting | Method | MNIST | | | CIFAR-10 | | |
|---|---|---|---|---|---|---|---|
| | | $Acc_{t=1}$ | $Acc_{t=5}$ | $Acc_{t=10}$ | $Acc_{t=1}$ | $Acc_{t=5}$ | $Acc_{t=10}$ |
| **Lifelong learning** | LWF | 20.17 | 26.49 | 32.34 | 20.00 | 20.00 | 20.00 |
| | EWC | 20.10 | 26.64 | 31.47 | 20.00 | 20.00 | 20.01 |
| | MAS | 20.10 | 26.64 | 31.47 | 20.00 | 20.00 | 20.00 |
| | GEM | 20.00 | 25.41 | 49.84 | 20.00 | 25.94 | 32.49 |
| | A-GEM | 20.00 | 20.94 | 22.47 | 20.00 | 20.38 | 20.07 |
| **Federated learning** | FedAvg | 43.09 | 51.76 | 59.50 | 19.94 | 20.84 | 23.88 |
| | FedProx | 49.12 | 56.17 | 60.28 | 21.70 | 23.74 | 26.32 |
| | SCAFFOLD | 48.04 | 56.14 | 64.44 | 21.09 | 23.78 | 26.18 |
| | FedAvg-FT | 27.36 | 50.27 | 59.10 | 20.00 | 20.34 | 23.02 |
| | FedProx-FT | 36.07 | 53.46 | 58.87 | 20.00 | 20.33 | 22.51 |
| | Ditto | 37.80 | 51.63 | 59.41 | 20.04 | 20.29 | 22.32 |
| | FedRep | 20.00 | 20.57 | 23.23 | 20.00 | 20.00 | 20.00 |
| | pFedGraph | 20.00 | 26.09 | 34.79 | 20.00 | 20.93 | 22.23 |
| **Decentralized optimization** | DSGD | 20.10 | 20.52 | 22.61 | 20.02 | 20.02 | 20.00 |
| | DSGT | 22.42 | 21.34 | 22.81 | 19.94 | 20.02 | 20.05 |
| | DFedAvgM | 36.91 | 51.37 | 54.91 | 19.97 | 21.30 | 22.38 |
| **Decentralized and lifelong-adaptive collaborative learning** | `DeLAMA-WC` | 37.15 | 78.92 | 95.54 | 30.96 | 57.81 | 71.46 |
| | `DeLAMA-WM` | **67.67** | 66.67 | 67.10 | 46.87 | 46.07 | 45.32 |
| | `DeLAMA` | **67.67** | **98.03** | **99.51** | **46.87** | **72.57** | **76.03** |

serving as the feature extractor:

$$\widehat{\mathcal{X}}_i^{(t)} = \phi_{\boldsymbol{\beta}}\left(\mathcal{X}_i^{(t)}\right), \tag{10a}$$

$$\mathbf{H}_i^{(t)} = \nabla^2_{\boldsymbol{\alpha}_i^{(t)}} \mathcal{L}\left(f_{\boldsymbol{\theta}_i^{(t)}}\left(\widehat{\mathcal{X}}_i^{(t)}\right), \mathcal{Y}_i^{(t)}\right). \tag{10b}$$

Compared to those iterations shown in **Algorithm** 1, the nonlinear transform $\phi_{\boldsymbol{\beta}}(\cdot)$ used in equation 10a targets to embed the input data $\mathcal{X}_i^{(t)}$ as $\widehat{\mathcal{X}}_i^{(t)}$, and uses this transformed data as the input of the model $f_{\boldsymbol{\theta}_i^{(t)}}(\cdot)$ in (10b). This approach offers two advantages. First, by incorporating the nonlinear backbone of the local learning task, the shared model $\boldsymbol{\theta}_i^{(t)}$ becomes more streamlined with a reduced number of parameters. This directly leads to a substantial reduction in communication overhead. Second, a simplified model $\boldsymbol{\theta}_i^{(t)}$ could also simplify the collaboration mechanism within $\boldsymbol{\Phi}_{\text{graph}}(\cdot)$ and $\boldsymbol{\Phi}_{\text{param}}(\cdot)$, making the collaboration process more efficient.

In summary, for equation 9a, we add a backbone with parameter $\beta$, for equation 9a equation 9b equation 9c we change the hyperparameters $\{\lambda_i\}_{i=1}^3$ to learnable parameters. Each iteration of the overall framework shown in equation 2 is unfolded into one specific layer of the deep neural network. The full inference network framework of `DeLAMA` is shown in **Figure** 2 and **Algorithm** 4, with learnable hyper-parameters highlighted in blue.

## 3.2 TRAINING DETAILS

Different from traditional training pipeline: training model parameters, then inference; here with algorithm unrolling, the training procedure of `DeLAMA` comprises two steps: the learning-to-learn step and the model learning step. This enables `DeLAMA` to capture an efficient collaboration strategy suitable for various tasks. First, in the learning-to-learn step, we train the hyperparameters $\gamma$ in the unrolled network $\mathcal{F}_\gamma(\cdot)$, optimizing a collaboration strategy. Second, in the model learning step, by executing a forward-pass of $\mathcal{F}_\gamma(\cdot)$ with a fixed hyperparameter, we obtain the output $\Theta^{(t)}$, representing the agents' model parameters. Through this, the advantage is enabling the collaboration mechanism to learn meta knowledge on how to collaborate under different task configurations. After training, each agent can infer using the model with parameter $\Theta^{(t)}$ as usual. See **Appendix** F for details.

Table 2: The system collaborative learning performance of different collaboration relationship learning methods. Here Oracle means we use graph structure where agents with the same observation configuration collaborate together. The evaluation metric contains both the graph-level GMSE metric against the Oracle structure and the system performance on the evaluation set.

| Data | Method | Decentralize | $GMSE_{t=1}$ | $GMSE_{t=3}$ | $GMSE_{t=5}$ | $Acc_{t=1}$ | $Acc_{t=3}$ | $Acc_{t=5}$ |
|---|---|---|---|---|---|---|---|---|
| **MNIST** | NS | ✓ | 0.9999 | 2.3562 | 1.3772 | 52.83 | 83.13 | 95.81 |
| | GLasso | ✗ | 1.7585 | 0.2728 | 0.1365 | 38.52 | 81.89 | 95.88 |
| | MTRL | ✗ | 2.1181 | 0.3179 | 0.1444 | 39.36 | 79.61 | 97.28 |
| | GL-LogDet | ✗ | 7.0086 | 0.9999 | 1.0000 | 44.93 | 65.43 | 79.93 |
| | L2G-PDS | ✗ | 0.5211 | 0.2970 | 0.1391 | 68.32 | 91.96 | 97.30 |
| | Oracle | - | - | - | - | 69.28 | 92.83 | 98.22 |
| | DeLAMA | ✓ | **0.4998** | **0.1968** | **0.0855** | **67.67** | **92.02** | **98.03** |
| **CIFAR-10** | NS | ✓ | 1.9756 | 4.0367 | 2.321 | 38.36 | 54.28 | 66.19 |
| | GLasso | ✗ | 14.3883 | 1.3897 | 0.3823 | 24.41 | 55.39 | 69.74 |
| | MTRL | ✗ | 375.2 | 2.0157 | 0.4352 | 25.36 | 53.23 | 64.88 |
| | GL-LogDet | ✗ | 1.0000 | 1.0033 | 1.0109 | 35.52 | 48.51 | 58.23 |
| | L2G-PDS | ✗ | 0.5067 | 0.3226 | 0.2065 | 45.63 | 65.03 | 72.06 |
| | Oracle | - | - | - | - | 48.21 | 68.51 | 74.16 |
| | DeLAMA | ✓ | **0.5948** | **0.3089** | **0.1475** | **46.87** | **65.19** | **72.57** |

## 4 EXPERIMENTS

We evaluate DeLAMA from four aspects, in **Section** 4.1, we delve into image classification tasks with more complex data forms; in **Section** 4.2, we apply DeLAMA to multi-robot mapping tasks, representing real scenarios' performance; In each aspect, performance is evaluated both quantitatively and qualitatively. We also conduct ablation studies on DeLAMA to highlight the significance of each module in the collaborative learning mechanism.

### 4.1 IMAGE CLASSIFICATION

**Task.** We consider a collaboration system with 6 collaborators. To encourage the collaboration behaviors between the agents, we divide the system of agents into two groups where each group's agents are all doing one 5-class classification task. We adopt the class incremental learning paradigm into our experiment, where each time agents randomly access purely one single class of data drawn from the five classes.

**Metric.** We adopt two metrics for evaluation: 1) the average prediction accuracy for each agent measured on the test set. 2) the graph mean square error shown in equation 25.

**Decentralized collaboration**

**Table** 1 presents the system comparison with life-long learning (Li and Hoiem, 2017; Kirkpatrick et al., 2017; Chaudhry et al., 2018), federated learning (McMahan et al., 2017; Li et al., 2020) and decentralized learning methods (Lian et al., 2017; Pu and Nedić, 2018; Hsu et al., 2019). The experimental result shows that i) our decentralized and lifelong collaboration approach could effectively remember previously learned training data, significantly outperforms classic lifelong learning approaches, showing the effectiveness of our lifelong learning mechanism; ii) compared with other multi-client and decentralized training approaches (McMahan et al., 2017; Ye et al., 2023; Hsu et al., 2019) that uses a static collaboration graph, our decentralized

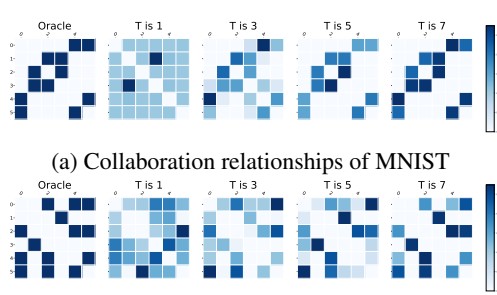

(a) Collaboration relationships of MNIST

(b) Collaboration relationships of CIFAR10

Figure 3: Collaboration relationships

collaboration significantly improves the classification accuracy, showing the effectiveness of our dynamic learning of collaboration graph; iii) adding our agent collaboration and remarkably improves the learning performance, especially when all the agents lack data at the beginning.

**Table** 2 presents the comparison with other classic relationship learning approaches (Meinshausen and Bühlmann, 2006; Banerjee et al., 2008; Pu et al., 2021), including centralized methods (Dong

Figure 4: The simulation result of robot mapping. `DeLAMA` outperforms other decentralized collaborative model training approaches. **Black:** empty space. **White:** occupied space.

et al., 2016; Liu et al., 2017) and decentralized approaches (Meinshausen and Bühlmann, 2006). We compare these approaches by substituting $\mathbf{\Phi}_{\text{graph}}(\cdot)$ of `DeLAMA` to these relational learning methods while preserving other parts of `DeLAMA` stay unchanged. The results show that i) `DeLAMA` outperforms classic relationship learning algorithms in classification tasks with a significant reduction in the graph structure inference error of **38.53%** and **28.57%**, reflecting that our method accurately learns the collaboration graph; ii) the accurate learning of collaboration graph further leads to the improvement of task performance (classification).

**Figure** 3 visualizes the time-evolving collaboration structures produced by `DeLAMA`. Compared with the oracle structure, the results on **MNIST** show that i) the collaboration relationships are evolving according to agents' observations.

**Lifelong adaptation**

We compare `DeLAMA-WC` with several lifelong learning approaches. **Table** 1 shows that even without collaboration, `DeLAMA-WC` outperforms standard lifelong learning approaches. For collaboration structures, we compare the system performance of `DeLAMA` with other static collaboration approaches such as federated learning and collaboration relationship learning. **Table** 1 and **Table** 2 reveals that i) `DeLAMA` has lifelong-adaptive capabilities rather than static collaboration approaches; ii) as timestamp increases, the learning performance improvement brought by our lifelong-adaptive learning enlarges.

## 4.2 MULTI-ROBOT MAPPING

In this section, we show that `DeLAMA` could be applied to multi-robot mapping scenarios. Specifically, we show that our decentralized collaborative mapping mechanism outperforms previous decentralized optimization and single-agent mappings.

**Task.** We consider a collaboratively robot 2D mapping task with 5 robots. The room structures are generated from the Cubi-Casa5k datasetKalervo et al. (2019). In this environment exploration task, robots are placed randomly in the room and set a manually predefined exploration trajectory. Each robot can have a local view of the environment based on LiDAR samples along the trajectory and cannot visit every part of the room. The goal of this task is to realize the efficient and accurate mapping for each agent through collaboration.

Table 3: The agents' average F1-score of the multi-robot mapping task. `DeLAMA` outperforms other decentralized collaborative model training approaches.

| Method | $\text{F1}_{t=1}$ | $\text{F1}_{t=6}$ | $\text{F1}_{t=12}$ |
|---|---|---|---|
| DSGD Lian et al. (2017) | 13.44 | 43.90 | 45.75 |
| DSGT Pu and Nedić (2018) | 38.32 | 38.31 | 38.31 |
| DiNNO Yu et al. (2021) | 14.73 | 20.34 | 25.57 |
| `DeLAMA-WC` | 68.50 | 72.66 | 74.92 |
| `DeLAMA` | **72.11** | **88.52** | **95.31** |

**Table** 3 shows the comparison with previous decentralized optimization methods Lian et al. (2017); Pu and Nedić (2018); Yu et al. (2021). We see that i) our collaboration learning strategy outperforms previous decentralized optimization methods and significantly improves the efficiency and performance of environmental exploration; ii) compared to `DeLAMA-WC` that is an individual agent mapping without collaboration, adding agent collaboration remarkably improves the exploration performance. **Figure** 4 illustrates that i) `DeLAMA` outperforms other decentralized model training

approaches, ii) compared to a single agent's limit perception range, multi-agent collaboration could discover a much more complete room structure. According to the visualization, the detection range and classification accuracy of the agent are improved with the help of multi-agent collaboration.

## 5 RELATED WORK

**Multi-Agent Communication.** Multi-agent communication enables agents to send messages to each other to realize inter-agent collaboration. The key part of multi-agent communication lies in the designing of inter-agent communication strategy (Singh et al., 2018). Traditional methods apply predefined features or heuristics (Qureshi and Terzopoulos, 2008; Li et al., 2010) to design communication protocols. More recently, learning-based methods have been proposed (Sukhbaatar et al., 2016), which train the communication protocols end-to-end under task performance supervision. These approaches are often applied in perception or decision-making, such as collaborative perception or multi-agent reinforcement learning (MARL). In collaborative perception, current methods mainly focus on designing communication strategies (Hu et al., 2022) or finding the exact time to share (Liu et al., 2020). In multi-agent reinforcement learning (Zhang et al., 2021), studies have explored various communication approaches, from simply sharing sensations, and policies to sharing other abstract data embeddings,like SARNet (Rangwala and Williams, 2020), TarMAC (Das et al., 2019), and IMAC (Wang et al., 2020),

**Lifelong Learning.** Lifelong learning (Wang et al., 2023) aims to train a continuously learning agent who can memorize previously learned tasks and quickly adapt to new training tasks. Optimizing model parameters directly each time may suffer from catastrophic forgetting (Goodfellow et al., 2013). To overcome this issue, current approaches can be categorized into three aspects: regularization-based, rehearsal-based, and architecture-increasing-based methods. Regularization-based methods like EWC (Kirkpatrick et al., 2017), PathInt (Zenke et al., 2017), and MAS (Aljundi et al., 2018) aim to estimate the important parts of model parameters and keep these parts changing slowly in model training, other methods like LWF (Li and Hoiem, 2017), target to keep previously learned data representations unchanged while learning new tasks. Rehearsal-based methods such as iCaRL (Rebuffi et al., 2017), EEIL (Castro et al., 2018), and GEM (Lopez-Paz and Ranzato, 2017) use previous tasks' data samples in real-time task learning, others like (Shin et al., 2017) utilize generative models to generate task data samples. TAMiL (Bhat et al., 2023) combines rehearsal methods with regularization approaches together. Architecture-increasing methods like (Rusu et al., 2016) try to utilize new task model parameters to prevent catastrophic forgetting, requiring relatively large memory capacities.

## 6 CONCLUSION

In this paper, we propose a novel decentralized lifelong-adaptive collaborative learning framework based on numerical optimization and algorithm unrolling, named `DeLAMA`. It enables multiple agents to efficiently detect collaboration relationships and adapt to ever-changing observations during individual model training. We validate the effectiveness of `DeLAMA` through extensive experiments on various real-world and simulated datasets. Experimental results show that `DeLAMA` achieves superior performances compared with other collaborative learning approaches.

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

## A   Task Setting

Considering a multi-agent system containing $N$ agents, each agent can train a dynamic machine-learning model for evolving tasks. At any given timestamp $t$, the $i$-th agent receives a training dataset $\mathcal{D}_i^{(t)} = \left\{ \mathcal{X}_i^{(t)}, \mathcal{Y}_i^{(t)} \right\}$, where $\mathcal{X}_i^{(t)}$ represents the data inputs and $\mathcal{Y}_i^{(t)}$ denotes the associated ground-truth labels. Based on this, the $i$-th agent updates its model parameters, denoted as $\boldsymbol{\theta}_i^{(t)}$, to effectively handle the test dataset $\widetilde{\mathcal{D}}_i^{(t)} = \left\{ \widetilde{\mathcal{X}}_i^{(t)}, \widetilde{\mathcal{Y}}_i^{(t)} \right\}$. It is worth noting that the test dataset $\widetilde{\mathcal{D}}_i^{(t)}$ shares the same distribution as the training dataset $\mathcal{D}_i^{(t)}$. The overall goal is to enhance inter-agents' collective learning performance. To achieve this, each agent can further 1) collaborate with other agents by sharing valuable information under a collaboration protocol, and 2) use its historical information in the learning process.

For decentralized multi-agent collaboration, information sharing is achieved by direct peer-to-peer communication with a time-varying communication graph $\mathbf{C}^{(t)} \in \{0, 1\}^{N \times N}$. Here, each node represents one agent and each $(i, j)$-th edge with $\mathbf{C}_{ij}^{(t)} = 1$ is a pairwise communication link indicating the message transmission between agent $i$, $j$ is enabled.

Given that the current learning task could relate to previous ones, each agent should be able to incorporate past task experiences into its current learning process. Let $\mathcal{M}_i^{(t)}$ be the learning memory of the $i$-th agent up to time $t$. To maintain and utilize the lifelong and adaptive learning ability, the storage space for the memory $\mathcal{M}_i^{(t)}$ of the entire learning task $\mathcal{T}_i$ is required to be finite, hence directly storing past training data is prohibited. Based on the constraint, agents will update the learning memory according to newly observed data $\mathcal{D}_i^{(t)}$ at each time, hence promoting lifelong adaptation.

Under the above conditions on information sharing and memory, decentralized and lifelong-adaptive collaborative learning aims to find an effective decentralized model training strategy. Denote the communication message sent from agent $i$ to agent $j$ at time $t$ as $m_{i \to j}^{(t)}$, where the information routing process could be multi-hop via the communication graph $\mathbf{C}^{(t)}$. The model parameter and memory updating rule are described as

$$\boldsymbol{\theta}_i^{(t)}, \mathcal{M}_i^{(t)} = \boldsymbol{\Phi}_i \left( \mathcal{D}_i^{(t)}, \{\boldsymbol{m}_{j \to i}^{(t)}\}_{j=1}^N, \mathcal{M}_i^{(t-1)} \big| \mathbf{C}^{(t)} \right), \tag{11}$$

where $\boldsymbol{\Phi}_i(\cdot)$ is the $i$-th agent's approach to update learning memory and model parameters according to previous individual learning memory $\mathcal{M}_i^{(t-1)}$, training data $\mathcal{D}_i^{(t)}$ and the messages $\{\boldsymbol{m}_{j \to i}^{(t)}\}_{j=1}^N$ sent from other agents.

Note that: i) the collaboration mechanism among the agents is decentralized, which means agents learn their model parameters based on individual observations and mutual collaboration without the management of a central server; ii) the collaboration system, including the agents' models and the collaboration strategies, is lifelong-adaptive to the dynamic scenarios, enabling agents to efficiently learn new knowledge while preserving knowledge from previously encountered ones; and iii) the sparse communication prior among agents based on the constraint $\mathbf{C}^{(t)}$ should be guaranteed, which means the inter-agent messages $\boldsymbol{m}_{j \to i}^{(t)}$ should adheres strictly to the communication graph constraint $\mathbf{C}^{(t)}$.

To evaluate the learning performance of the $i$-th agent at timestamp $t$, we use the performance accumulation as the evaluation metric:

$$L_i^{(t)} = \frac{1}{t} \sum_{k=1}^{t} \mathcal{L} \left( f_{\boldsymbol{\theta}_i^{(k)}}(\widetilde{\mathcal{X}}_i^{(k)}), \widetilde{\mathcal{Y}}_i^{(k)} \right),$$

where $\widetilde{\mathcal{X}}_i^{(t)}$ is the test data and $\widetilde{\mathcal{Y}}_i^{(t)}$ is the associated test labels. $\mathcal{L}(\cdot)$ is the metric function corresponding to the task. The overall performance of $N$ agents is $L^{(t)} = L_i^{(t)}/N$. The task objective is to maximize $L^{(t)}$ by designing, optimizing the communication protocol of sharing messages $\boldsymbol{m}_{j \to i}^{(t)}$ and the model updating approach $\boldsymbol{\Phi}_i(\cdot)$ based on the observed data $\mathcal{D}_{1:N}^{(t)} = \left\{ \mathcal{D}_i^{(t)} \right\}_{i=1}^N$ at each timestamp.

## B    OPTIMIZATION PROBLEM

We consider solving this decentralized and lifelong-adaptive collaboration task shown in **Section** A from a probabilistic perspective. According to the task setting described in **Section** A, suppose the previous encountered datasets of agent $i$ until time $t$ is written as $\mathcal{D}_{\mathcal{T}_i}^t = \left\{ \mathcal{D}_i^{(k)} \big| 1 \leq k \leq t \right\}$, where $\mathcal{T}_i$ represents the unique learning task's experience index of the $i$-th agent. Each time $t$ the collaboration system will estimate a group of model parameters $\Theta^{(t)} = \left\{ \boldsymbol{\theta}_i^{(t)} \big| 1 \leq i \leq N \right\}$ given the group of the datasets $\mathcal{D}_{\mathcal{T}_{1:N}}^t$, where $\mathcal{D}_{\mathcal{T}_{1:N}}^t = \left\{ \mathcal{D}_{\mathcal{T}_i}^t \big| 1 \leq i \leq N \right\}$ is the full dataset for agent $1 \leq i \leq N$. Since this is a decentralized lifelong adaptive collaboration scheme where direct sharing training data is forbidden, each agent at time $t$ can only access to $p\left( \boldsymbol{\theta}_i^{(t)} \big| \mathcal{D}_i^{(t)} \right)$ or $p\left( \mathcal{D}_i^{(t)} \big| \boldsymbol{\theta}_i^{(t)} \right)$. Hence simply using the likelihood function $p\left( \mathcal{D}_{\mathcal{T}_{1:N}}^t \big| \Theta^{(t)} \right)$ is impractical due to complex inter-agent correlations between agents' model parameters. Instead, we consider the Bayesian learning paradigm by maximizing the posterior distribution of $\Theta^{(t)}$ as $p\left( \Theta^{(t)} \big| \mathcal{D}_{\mathcal{T}_{1:N}}^t \right)$ after knowing the datasets $\mathcal{D}_{\mathcal{T}_{1:N}}^t$. The inter-agents' correlations are described by the prior distribution of $\Theta^{(t)}$. Specifically, according to the Bayes rule, the posterior distribution can be decomposed into the following two parts:

$$p\left( \Theta^{(t)} \big| \mathcal{D}_{\mathcal{T}_{1:N}}^t \right) \propto p\left( \mathcal{D}_{\mathcal{T}_{1:N}}^t \big| \Theta^{(t)} \right) p\left( \Theta^{(t)} \right), \tag{12}$$

where the first probability is the likelihood of the agents model parameters on their datasets $\mathcal{D}_{\mathcal{T}_{1:N}}$, and the second probability $p\left( \Theta^{(t)} \right)$ is the prior distribution of $\Theta^{(t)}$.

**Likelihood probability** $p\left( \mathcal{D}_{\mathcal{T}_{1:N}}^t \big| \Theta^{(t)} \right)$ **:** Since the training data arrives independently among the agents at each time $t$, we assume that the data batches $\mathcal{D}_{\mathcal{T}_i}^{(j)}$ satisfy conditional independency on both agent-level for $1 \leq i \leq N$ and time-level for $1 \leq k \leq t$ shown in the following equation:

$$\textbf{Agent: } p\left( \mathcal{D}_{\mathcal{T}_1}^t, \mathcal{D}_{\mathcal{T}_2}^t, \ldots, \mathcal{D}_{\mathcal{T}_N}^t \big| \Theta^{(t)} \right) = \prod_{i=1}^N p\left( \mathcal{D}_{\mathcal{T}_i}^t \big| \Theta^{(t)} \right),$$

$$\textbf{Time: } p\left( \mathcal{D}_i^{(1)}, \mathcal{D}_i^{(2)}, \ldots, \mathcal{D}_i^{(t)} \big| \boldsymbol{\theta}_i^{(t)} \right) = \prod_{k=1}^t p\left( \mathcal{D}_i^{(k)} \big| \boldsymbol{\theta}_i^{(t)} \right), \tag{13}$$

where the first equation represents agent-level conditional independence and the second denotes time-level conditional independence, standing for the task data $\mathcal{D}_{\mathcal{T}_{1:N}}^t$ arrives independently among the agents as time progresses. Thus we decompose the likelihood $p\left( \mathcal{D}_{\mathcal{T}_{1:N}}^t \big| \Theta^{(t)} \right)$ as:

$$p\left( \mathcal{D}_{\mathcal{T}_{1:N}}^t \big| \boldsymbol{\theta}_i^{(t)} \right) = \prod_{i=1}^N \prod_{k=1}^t p\left( \mathcal{D}_i^{(k)} \big| \boldsymbol{\theta}_i^{(t)} \right), \tag{14}$$

where $p\left( \mathcal{D}_i^{(j)} \big| \boldsymbol{\theta}_i^{(t)} \right)$ is the likelihood corresponding to the discriminative model for each agent. In equation 14, the decomposition follows from both the agent-level and time-level conditional independence as shown in equation 13.

**Prior probability** $p\left( \Theta^{(t)} \right)$ **:** To model the prior information of $\Theta^{(t)}$, we consider the probabilistic approach (Dong et al., 2016) by parameterizing this prior distribution with pairwise correlations among multiple agents' model parameters. Specifically, unlike the agent-level independence of the training data, these correlations between agents describe the relationships between model parameters, which correspond to a graph $\mathcal{G}^{(t)} = \{\mathcal{V}, \mathcal{E}^{(t)}\}$. Here $\mathcal{V}$ is the node set of the agents and $\mathcal{E}^{(t)}$ is the edge set at timestamp $t$. The adjacency matrix of graph $\mathcal{G}^{(t)}$ is defined as $\mathbf{W}^{(t)} \in \mathbb{R}^{N \times N}$, indicating the collaboration relationships among the agents, used to guide the collaboration behaviors. Note that the collaboration relationship $\mathbf{W}^{(t)}$ is restricted by the communication graph $\mathbf{C}^{(t)}$ as its substructure. The weight of $\mathbf{W}^{(t)}$ represents the model similarity between agents. Assuming that similar model parameters promote agents to collaborate, where more similar model parameters correspond to stronger collaboration relations, the prior distribution of model parameters $\Theta^{(t)}$ given collaboration

---

**Algorithm 1** Local learning $\Phi_{\text{local}}(\cdot)$

---

**Input data:** $\mathcal{D}_{1:N}^{(t)} = \left\{ \mathcal{D}_i^{(t)} \big| \mathcal{D}_i^{(t)} = \left( \mathbf{X}_i^{(t)}, \mathbf{Y}_i^{(t)} \right), 1 \leq i \leq N \right\}$

**for** $i = 1, 2, \ldots, N$ **(parallel) do**

    Initialize expansion point $\boldsymbol{\alpha}_i^{(t)}$

    $\mathbf{H}_i^{(t)} = \nabla_{\boldsymbol{\alpha}_i^{(t)}}^2 \mathcal{L} \left( f_{\boldsymbol{\theta}_i^{(t)}} \left( \mathbf{X}_i^{(t)} \right), \mathbf{Y}_i^{(t)} \right)$ { Calculate Hessian}

    $\mathbf{A}_i^{(t)} = \left( (t-1)\mathbf{A}_i^{(t-1)} + \mathbf{H}_i^{(t)} \right) / t$   {Update history $\mathbf{A}_i^{(t)}$}

    $\boldsymbol{b}_i^{(t)} = \left( (t-1)\boldsymbol{b}_i^{(t-1)} + \mathbf{H}_i^{(t)}\boldsymbol{\alpha}_i^{(t)} \right) / t$   {Update history $\boldsymbol{b}_i^{(t)}$}

    $\boldsymbol{\theta}_i^{(t,0)} = \left[ \mathbf{A}_i^{(t)} + 2\lambda_1 \mathbf{I} \right]^{-1} \boldsymbol{b}_i^{(t)}$   {Calculate parameter $\boldsymbol{\theta}_i^{(t,0)}$}

**end for**

---

relationship $\mathbf{W}^{(t)}$ and the communication graph $\mathbf{C}^{(t)}$ are described by:

$$
\begin{aligned}
- \log p \left( \boldsymbol{\Theta}^{(t)} \right) &\propto \sum_{1 \leq i,j \leq N} \mathbf{W}_{ij}^{(t)} \left\| \boldsymbol{\theta}_i^{(t)} - \boldsymbol{\theta}_j^{(t)} \right\|_2^2 \\
&= \mathbf{tr} \left( \boldsymbol{\Theta}^{(t)\top} \mathbf{L}^{(t)} \boldsymbol{\Theta}^{(t)} \right),
\end{aligned}
\tag{15}
$$

where $\mathbf{W}_{ij}^{(t)} = 0$ if $\mathbf{C}_{ij}^{(t)} = 0$. Here $\mathbf{tr} \left( \boldsymbol{\Theta}^{(t)\top} \mathbf{L}^{(t)} \boldsymbol{\Theta}^{(t)} \right)$ is the smoothness of $\boldsymbol{\Theta}^{(t)}$ viewed as functions on the graph and $\mathbf{L}^{(t)}$ is the combinatorial graph laplacian defined as $\mathbf{L}^{(t)} = \mathbf{D}^{(t)} - \mathbf{W}^{(t)}$ at time $t$. $\mathbf{D}^{(t)}$ is a diagonal matrix with its $i$-th entry representing the degree of the $i$-th node.

Taking the likelihood probability shown in equation 14 and the prior probability shown in equation 15 to the original posterior probability $p \left( \boldsymbol{\Theta}^{(t)} \big| \mathcal{D}_{\mathcal{T}_{1:N}}^t \right)$ in equation 12, we obtain a decomposed version of probability:

$$
p \left( \boldsymbol{\Theta}^{(t)} \big| \mathcal{D}_{\mathcal{T}_{1:N}}^t \right) \propto \prod_{i=1}^N \prod_{k=1}^t p \left( \mathcal{D}_i^{(k)} \big| \boldsymbol{\theta}_i^{(t)} \right) p \left( \boldsymbol{\Theta}^{(t)} \right).
\tag{16}
$$

Thus we can reformulate equation 12 as the following non-constraint optimization problem:

$$
\min_{\boldsymbol{\theta}_i^{(t)}} \sum_{i=1}^N \sum_{k=1}^t - \log p \left( \mathcal{D}_i^{(k)} \big| \boldsymbol{\theta}_i^{(t)} \right) + \lambda \, \mathbf{tr} \left( \boldsymbol{\Theta}^{(t)\top} \mathbf{L}^{(t)} \boldsymbol{\Theta}^{(t)} \right),
\tag{17}
$$

where the prior probability distribution of $\boldsymbol{\Theta}^{(t)}$ known as $p \left( \boldsymbol{\Theta}^{(t)} \right)$ corresponds to the graph smoothness regularization $\mathbf{tr} \left( \boldsymbol{\Theta}^{(t)\top} \mathbf{L}^{(t)} \boldsymbol{\Theta}^{(t)} \right)$.

## C  LOCAL LEARNING

### C.1  ALGORITHM

### C.2  SOLUTION

Specifically, recall that the definition of $\ell^k \left( \boldsymbol{\theta}_i^{(t)} \right) = - \log p \left( \mathcal{D}_i^{(k)} \big| \boldsymbol{\theta}_i^{(t)} \right)$ shown in equation 1 is the loss function correspond to $\mathcal{D}_i^{(k)}$ for model parameter $\boldsymbol{\theta}_i^{(t)}$. Suppose we perform direct Taylor expansion at one point $\boldsymbol{\alpha}_i^{(k)}$ of $\ell^k \left( \boldsymbol{\theta}_i^{(t)} \right)$. Then, the accumulated loss $\mathcal{L}_i^{(t)} \left( \boldsymbol{\theta}_i^{(t)} \right) = \frac{1}{t} \sum_{k=1}^t \ell^k \left( \boldsymbol{\theta}_i^{(t)} \right)$ is approximated by

$$
\begin{aligned}
\mathcal{L}_i^{(t)} \left( \boldsymbol{\theta}_i^{(t)} \right) &\approx \frac{1}{t} \sum_{k=1}^t \frac{1}{2} \left( \boldsymbol{\theta}_i^{(t)} - \boldsymbol{\alpha}_i^{(k)} \right)^\top \mathbf{H}_i^{(k)} \left( \boldsymbol{\theta}_i^{(t)} - \boldsymbol{\alpha}_i^{(k)} \right) \\
&\quad + \left( \boldsymbol{\theta}_i^{(t)} - \boldsymbol{\alpha}_i^{(k)} \right)^\top \boldsymbol{g}_i^{(k)} + \ell^k \left( \boldsymbol{\alpha}_i^{(k)} \right),
\end{aligned}
\tag{18}
$$

---

**Algorithm 2** Collaborative relational inference $\mathbf{\Phi}_{\text{graph}}(\cdot)$

---

**Input:** $\mathbf{\Theta}^{(t)} = \left( \boldsymbol{\theta}_1^{(t)}, \ldots, \boldsymbol{\theta}_n^{(t)} \right)$
**for** $k = 0, 1, 2, \ldots, M$ **do**
  **for** $i = 1, 2, \ldots, N$ (**parallel**) **do**
    **Initialize:** $\boldsymbol{z}_i^0 = \mathbf{0}, 1 \leq i \leq N$
    $\boldsymbol{u}_i^k = -\left( \lambda_2 \boldsymbol{d}_i^{(t)} + \boldsymbol{z}_i^{k-1} \mathbf{1} \right) / (2\lambda_3)$
    $\boldsymbol{w}_i^{(t),k}, \boldsymbol{w}_i^{\prime(t),k}, \boldsymbol{w}_i^{\prime\prime(t),k} = h\left( \boldsymbol{u}_i^k \right), h'\left( \boldsymbol{u}_i^k \right), h''\left( \boldsymbol{u}_i^k \right)$
    $\boldsymbol{x}_i^k = \boldsymbol{u}_i^{k\top} \boldsymbol{w}_i^{\prime(t),k} + \boldsymbol{w}_i^{(t),k\top} \left( \mathbf{1} - \boldsymbol{w}_i^{\prime\prime(t),k} \right)$
    $\boldsymbol{s}_i^k = \left( 2\mathbf{1} - \boldsymbol{w}_i^{\prime(t),k} \right)^{\top} \boldsymbol{w}_i^{\prime(t),k}$
    $\boldsymbol{r}_i^k = \left( \boldsymbol{u}_i^k - \boldsymbol{w}_i^{\prime(t),k} \right)^{\top} \boldsymbol{w}_i^{\prime\prime(t),k}$
    $\boldsymbol{y}_i^k = \boldsymbol{s}_i^k + \boldsymbol{r}_i^k$
    **Gather** $\boldsymbol{x}_j^k, \boldsymbol{y}_j^k, 1 \leq j \leq N$ via $\mathbf{C}^{(t)}$ to Agent $i$
    $\boldsymbol{p}_i^k = \sum_{j=1}^{N} \boldsymbol{x}_j^k - \boldsymbol{m}$
    $\boldsymbol{q}_i^k = -1/\lambda_3 \sum_{j=1}^{N} \boldsymbol{y}_j^k$
    $\boldsymbol{z}_i^k = \boldsymbol{z}_i^{k-1} - \boldsymbol{p}_i^k / \boldsymbol{q}_i^k$ {Dual update $z_i$}
  **end for**
**end for**
**Output:** $\mathbf{W}^{(t)} \Leftarrow \left( \boldsymbol{w}_1^{(t),M}, \ldots, \boldsymbol{w}_N^{(t),M} \right)$

---

where $\mathbf{H}_i^{(k)}$ is the Hessian of $\ell^k \left( \boldsymbol{\theta}_i^{(t)} \right)$ at $\boldsymbol{\alpha}_i^{(k)}$, and $\boldsymbol{g}_i^{(k)}$ is the corresponding gradient at the same point. Thus the gradient of $\boldsymbol{\theta}_i^{(t)}$ at time $t$ can be approximated as

$$
\begin{aligned}
\frac{\partial \mathcal{L}_i^{(t)} \left( \boldsymbol{\theta}_i^{(t)} \right)}{\partial \boldsymbol{\theta}_i^{(t)}} &= \frac{1}{t} \sum_{k=1}^{t} \mathbf{H}_i^{(k)} \left( \boldsymbol{\theta}_i^{(t)} - \boldsymbol{\alpha}_i^{(k)} \right) + \boldsymbol{g}_i^{(k)} \\
&= \mathbf{A}^{(t)} \boldsymbol{\theta}_i^{(t)} - \boldsymbol{b}_i^{(t)},
\end{aligned}
\tag{19}
$$

where $\mathbf{A}_i^{(t)}, \boldsymbol{b}_i^{(t)}$ are two intermediate variables that can be understood as the linear summation of historical variables.

## D  COLLABORATIVE RELATIONAL INFERENCE

### D.1  ALGORITHM

### D.2  SOLUTION

Here we aim to solve the collaborative relational inference problem shown in equation 4. To enable the decentralized calculation of the collaboration relationships among the agents, we aim to split the graph adjacency matrix $\mathbf{W}^{(t)}$ into $N$ blocks. Specifically, similar to the reformulation trick shown in (Pu et al., 2021), we define the block correspond to the $i$-th agent be $\boldsymbol{w}_i^{(t)} \in \mathbb{R}^{N-1}$ as the $i$-th row of the adjacency matrix $\mathbf{W}^{(t)}$ except the $i$-th element. Define the parameter distance vector of the $i$-th agent as $\boldsymbol{d}_i^{(t)} \in \mathbb{R}^{N-1}$. Due to the existence of communication constraint $\mathbf{C}^{(t)}$, we extend the definition of $\boldsymbol{d}_i^{(t)}$ by $\boldsymbol{d}_{ij}^{(t)} = \left\| \boldsymbol{\theta}_i^{(t)} - \boldsymbol{\theta}_j^{(t)} \right\|_2^2 / \mathbf{C}_{ij}^{(t)}$. Then, an equivalent form of **Problem** equation 4 is

$$
\min_{\left( \boldsymbol{w}_1^{(t)}, \ldots, \boldsymbol{w}_N^{(t)} \right)} \sum_{i=1}^{N} \lambda_2 \boldsymbol{w}_i^{(t)\top} \boldsymbol{d}_i^{(t)} + \lambda_3 \left\| \boldsymbol{w}_i^{(t)} \right\|_2^2 + \mathcal{I}_{x>0} \left( \boldsymbol{w}_i^{(t)} \right)
$$

$$
\text{s.t.} \sum_{i=1}^{N} \mathbf{1}^{\top} \boldsymbol{w}_i^{(t)} = \boldsymbol{m}.
\tag{20}
$$

---

**Algorithm 3** Lifelong model update $\Phi_{\text{param}}(\cdot)$

---

**Initialize:** $\left( \mathbf{A}_i^{(t)}, \boldsymbol{b}_i^{(t)} \right), 1 \le i \le n$

**Input:** $\left( \boldsymbol{\theta}_1^{(t),0}, \ldots, \boldsymbol{\theta}_N^{(t),0} \right), \mathbf{W}^{(t)}$

**for** $k = 1, 2, \ldots, M$ **do**
    **for** $i = 1, 2, \ldots, N$ (**parallel**) **do**
        send local model $\boldsymbol{\theta}_i^{(t),k-1}$ to agents in $\mathcal{N}(i)$
        receive models $\left\{ \boldsymbol{\theta}_j^{(t),k-1} \big| j \in \mathcal{N}(i) \right\}$ from agents in $\mathcal{N}(i)$
        $\widetilde{\boldsymbol{\theta}}_i^{(t),k} \Leftarrow \boldsymbol{b}_i^{(t)} + 4\lambda_2 \sum_{j \in \mathcal{N}(i)} \mathbf{W}_{ij}^{(t)} \boldsymbol{\theta}_j^{(t),k-1}$
        $\mathbf{B}_i^{(t)} \Leftarrow \mathbf{A}_i^{(t)} + \left( 2\lambda_1 + 4\lambda_2 \mathbf{D}_i^{(t)} \right) \mathbf{I}$
        $\boldsymbol{\theta}_i^{(t),k} \Leftarrow \left( \mathbf{B}_i^{(t)} \right)^{-1} \widetilde{\boldsymbol{\theta}}_i^{(t),k}$
    **end for**
**end for**
**Output:** $\left( \boldsymbol{\theta}_1^{(t)}, \ldots, \boldsymbol{\theta}_N^{(t)} \right) \Leftarrow \left( \boldsymbol{\theta}_1^{(t,M)}, \ldots, \boldsymbol{\theta}_N^{(t,M)} \right)$

---

To solve this optimization problem, we use the Lagrange multiplier method to analyze the KKT conditions of the solution $\boldsymbol{w}_i^{(t)}$s. This corresponds to solving the following single-variable equation

$$\sum_{i=1}^N \mathbf{1}^\top \mathbf{ReLU} \left( -\frac{\lambda_2 \boldsymbol{d}_i^{(t)} + z\mathbf{1}}{2\lambda_3} \right) = \boldsymbol{m}, \tag{21}$$

where $z$ corresponds to the Lagrange multiplier of the $L_1$ equality constraint and $\mathbf{1} \in \mathbb{R}^{N-1}$.

To find the root $z$ of this non-smooth function, we propose an approximation method that tries to use a twice differentiable function $h(x) = (\sqrt{x^2 + b} + x)/2$ to approximate **ReLU**. This approximation provides two advantages: i) it yields acceptable solutions that are easily derived; and ii) it enhances efficiency, requiring fewer iterations when employing second-order optimization methods. We leverage the Newton iterations to equation 21, which could converge to the target solution much faster than state-of-the-art graph learning algorithms (Pu et al., 2021).

# E   Lifelong Model Update

## E.1   Algorithm

## E.2   Solution

Here we aim to solve the lifelong model update problem in equation 6. To realize the decentralized optimization of $\boldsymbol{\Theta}^{(t)}$, we aim to optimize the problem in $N$ blocks $\boldsymbol{\Theta}^{(t)} = \left[ \boldsymbol{\theta}_1^{(t)}, \ldots, \boldsymbol{\theta}_N^{(t)} \right]$. For each block of the model parameters $\boldsymbol{\theta}_i^{(t)}$, the first-order-condition of the problem shown in equation 6 can be described as a linear equation by taking the gradient approximation equation 19 into consideration:

$$\mathbf{A}_i^{(t)} \boldsymbol{\theta}_i^{(t)} - \boldsymbol{b}_i^{(t)}$$
$$= -2\lambda_1 \boldsymbol{\theta}_i^{(t)} - 2\lambda_2 \sum_{j=1}^N \left( \mathbf{W}_{ij}^{(t)} + \mathbf{W}_{ji}^{(t)} \right) \left( \boldsymbol{\theta}_i^{(t)} - \boldsymbol{\theta}_j^{(t)} \right). \tag{22}$$

Hence optimizing $\boldsymbol{\Theta}^{(t)}$ corresponds to solving a linear system with $N$ first-order conditions. To obtain the decentralized computation and reduce the computational overhead, we consider Jacobi-iteration Golub and Van Loan (2013) based approaches. Specifically, assuming the symmetry of the adjacency matrix $\mathbf{W}^{(t)}$, we rewrite equation 22 as

$$\left( \mathbf{A}_i^{(t)} + 2\lambda_1 \mathbf{I} + 4\lambda_2 \mathbf{D}_i^{(t)} \right) \boldsymbol{\theta}_i^{(t)} = \boldsymbol{b}_i^{(t)} + 4\lambda_2 \sum_{j=1}^N \mathbf{W}_{ij}^{(t)} \boldsymbol{\theta}_j^{(t)},$$

---

**Algorithm 4** DeLAMA

---

**Network parameter:** $\gamma = \left(\beta, \{\lambda_i\}_{i=1}^3\right)$

**for** $t = 1, 2, \ldots T$ **do**

  **Input:** $\mathcal{D}_{1:N}^{(t)} = \left\{\mathcal{D}_i^{(t)} \big| \mathcal{D}_i^{(t)} = \left(\mathcal{X}_i^{(t)}, \mathcal{Y}_i^{(t)}\right), 1 \leq i \leq N\right\}$

  /∗ Unfold the iterations of $\boldsymbol{\Phi}_{\text{local}}(\cdot)$                                                                       ∗/

  **for** $i = 1, 2, \ldots, N$ **parallel do**

    $\widehat{\mathcal{X}}_i^{(t)} = \phi_{\boldsymbol{\beta}}\left(\mathcal{X}_i^{(t)}\right)$

    **Initialize expansion point** $\boldsymbol{\alpha}_i^{(t)}$

    $\mathbf{H}_i^{(t)} = \nabla_{\boldsymbol{\alpha}_i^{(t)}}^2 \mathcal{L}\left(f_{\boldsymbol{\theta}_i^{(t)}}\left(\widehat{\mathcal{X}}_i^{(t)}\right), \mathcal{Y}_i^{(t)}\right)$ {Hessian}

    $\mathbf{A}_i^{(t)} = \left((t-1)\mathbf{A}_i^{(t-1)} + \mathbf{H}_i^{(t)}\right)/t$   {Update $\mathbf{A}_i^{(t)}$}

    $\boldsymbol{b}_i^{(t)} = \left((t-1)\boldsymbol{b}_i^{(t-1)} + \mathbf{H}_i^{(t)}\boldsymbol{\alpha}_i^{(t)}\right)/t$   {Update $\boldsymbol{b}_i^{(t)}$}

    $\boldsymbol{\theta}_i^{(t,0)} = \left[\mathbf{A}_i^{(t)} + 2\lambda_1\mathbf{I}\right]^{-1} \boldsymbol{b}_i^{(t)}$     {Calculate $\boldsymbol{\theta}_i^{(t,0)}$}

  **end for**

  /∗ Unfold the iteration of $\boldsymbol{\Phi}_{\text{graph}}(\cdot)$                                                                        ∗/

  **for** $k = 0, 1, 2, \ldots, M_1$ **do**

    **for** $i = 1, 2, \ldots, N$ **parallel do**

      **Initialize:** $\boldsymbol{z}_i^0, 1 \leq i \leq N$

      $\boldsymbol{u}_i^k = -\left(\lambda_2 \boldsymbol{d}_i^{(t)} + \boldsymbol{z}_i^{k-1}\mathbf{1}\right)/(2\lambda_3)$

      $\boldsymbol{w}_i^{(t),k}, \boldsymbol{w}_i'^{(t),k}, \boldsymbol{w}_i''^{(t),k} = h\left(\boldsymbol{u}_i^k\right), h'\left(\boldsymbol{u}_i^k\right), h''\left(\boldsymbol{u}_i^k\right)$

      $\boldsymbol{x}_i^k = \boldsymbol{u}_i^{k\top}\boldsymbol{w}_i'^{(t),k} + \boldsymbol{w}_i^{(t),k\top}\left(\mathbf{1} - \boldsymbol{w}_i''^{(t),k}\right)$

      $\boldsymbol{s}_i^k = \left(2\mathbf{1} - \boldsymbol{w}_i'^{(t),k}\right)^\top \boldsymbol{w}_i'^{(t),k}$

      $\boldsymbol{r}_i^k = \left(\boldsymbol{u}_i^k - \boldsymbol{w}_i'^{(t),k}\right)^\top \boldsymbol{w}_i''^{(t),k}$

      $\boldsymbol{y}_i^k = \boldsymbol{s}_i^k + \boldsymbol{r}_i^k$

      **Gather** $\boldsymbol{x}_j^k, \boldsymbol{y}_j^k, 1 \leq j \leq N$ via $\mathbf{G}^{(t)}$ to Agent $i$

      $\boldsymbol{p}_i^k = \sum_{j=1}^N \boldsymbol{x}_j^k - \boldsymbol{m}$

      $\boldsymbol{q}_i^k = -1/\lambda_3 \sum_{j=1}^N \boldsymbol{y}_j^k$

      $\boldsymbol{z}_i^k = \boldsymbol{z}_i^{k-1} - \boldsymbol{p}_i^k/\boldsymbol{q}_i^k$ {Dual update $z_i$}

    **end for**

  **end for**

  $\mathbf{W}^{(t)} = \left(\boldsymbol{w}_1^{M_1}, \ldots, \boldsymbol{w}_N^{M_1}\right)$

  /∗ Unfold the iteration of $\boldsymbol{\Phi}_{\text{param}}(\cdot)$                                                                       ∗/

  **for** $k = 1, 2, \ldots, M_2$ **do**

    **for** $i = 1, 2, \ldots, N$ **parallel do**

      /∗ Aggregate model parameters according to $\mathbf{W}^{(t)}$                                 ∗/

      $\widetilde{\boldsymbol{\theta}}_i^{(t),k} \Leftarrow \boldsymbol{b}_i^{(t)} + 4\lambda_2 \sum_{j \in \mathcal{N}(i)} \mathbf{W}_{ij}^{(t)} \boldsymbol{\theta}_j^{(t),k-1}$

      $\mathbf{B}_i^{(t)} \Leftarrow \mathbf{A}_i^{(t)} + \left(2\lambda_1 + 4\lambda_2\mathbf{D}_i^{(t)}\right)\mathbf{I}$

      $\boldsymbol{\theta}_i^{(t),k} \Leftarrow \left(\mathbf{B}_i^{(t)}\right)^{-1} \widetilde{\boldsymbol{\theta}}_i^{(t),k}$

    **end for**

  **end for**

  **Output:** $\left(\boldsymbol{\theta}_1^{(t,M_2)}, \ldots, \boldsymbol{\theta}_N^{(t,M_2)}\right)$

**end for**

---

where $\mathbf{D}_i^{(t)}$ is the degree of the $i$-th agent of the collaboration relation $\mathbf{W}^{(t)}$.

# F   UNROLLING

## F.1   ALGORITHM

### F.2 LEARNING TO LEARN THE COLLABORATION STRATEGY

We leverage the learning-to-learn paradigm to the training process of network $\mathcal{F}_\gamma(\cdot)$. Specifically, to train an efficient parameter learning network $\mathcal{F}_\gamma(\cdot)$, we use a bunch of training tasks, each represented by $\mathcal{T}_{1:N}^{train}$ where $\mathcal{T}_i^{train} \sim \mathcal{P}(\mathcal{T}_i)$ for $1 \leq i \leq N$, to provide sufficient examples and supervisions to tune the parameters $\gamma$. Here $\mathcal{P}(\mathcal{T}_i)$ represents the task distribution of the $i$-th agent, which could be used to sample different training task sequences. After this training, the collaboration strategy can be applied to tasks $\mathcal{T}_i$ following the same distribution.

**Network supervision.** The network supervision for one task $\mathcal{T}_{1:N}^{train}$ is the expected system average task loss:

$$\ell\left(\mathcal{T}_{1:N}^{train}\big|\gamma\right) = \frac{1}{Nt}\sum_{i=1}^{N}\sum_{k=1}^{t}\mathbb{E}_{(\mathcal{X}_i,\mathcal{Y}_i)\sim\mathcal{T}_i^{train}}\left[\mathcal{L}\left(f_{\boldsymbol{\theta}_i^{(k)}}(\mathcal{X}_i),\mathcal{Y}_i\right)\right],$$

where $\mathcal{L}(\cdot)$ is the task-specific loss and $f_{\boldsymbol{\theta}_i^{(k)}}(\cdot)$ is the learned model with parameter $\boldsymbol{\theta}_i^{(k)}$ generated from the network $\mathcal{F}_\gamma(\cdot)$ at timestamp $k$.

**Network training.** Training network $\mathcal{F}_\gamma(\cdot)$ corresponds to figuring out the best mechanism to collaborate and learn model parameters $\boldsymbol{\Theta}^{(t)}$ according to the task $\mathcal{T}_{1:N}^{train}$. Mathematically, this means minimizing the expected supervision of the training tasks $\mathcal{T}_{1:N}^{train}$ according to the rule:

$$\gamma^* = \arg\min_\gamma \mathbb{E}_{\mathcal{T}_{1:N}^{train}\sim\mathcal{P}(\mathcal{T}_{1:N})}\ell\left(\mathcal{T}_{1:N}^{train}\big|\gamma\right),$$

**Network evaluation.** Once trained, the optimal $\mathcal{F}_{\gamma^*}(\cdot)$ implicitly carries prior collaboration strategies that can be applied to new task learning configurations following the distribution $\mathcal{P}(\mathcal{T}_{1:N})$. In the evaluation phase, the evaluation metric is the expected network supervision loss on the test tasks $\mathcal{T}_{1:N}^{test}$:

$$L = \mathbb{E}_{\mathcal{T}_{1:N}^{test}\sim\mathcal{P}(\mathcal{T}_{1:N})}\ell\left(\mathcal{T}_{1:N}^{test}\big|\gamma^*\right),$$

where $\gamma^*$ is the optimal parameter of $\mathcal{F}_\gamma(\cdot)$.

This training approach can be understood as utilizing plenty of learning tasks $\mathcal{T}_{1:N}^{train}$ as examples to supervise the network how to learn model parameters, and subsequently applying the acquired knowledge to new learning tasks.

#### F.2.1 AGENT'S MODEL LEARNING

Learning the model parameters $\boldsymbol{\Theta}^{(t)}$ corresponds to the forward process of the unrolled network $\mathcal{F}_{\gamma^*}(\cdot)$ defined in equation 8, which takes the training data $\mathcal{D}_{1:N}^{(t)}$ as input and outputs $\boldsymbol{\Theta}^{(t)}$. During the forward pass of $\mathcal{F}_{\gamma^*}(\cdot)$, the finite learning memory $\mathcal{M}_{1:N}^{(t-1)}$ is recurrently updated, dynamically adapting to future observed tasks. Compared to the optimization procedure $\mathcal{F}(\cdot)$ defined in equation 2, the number of communication steps of $\Phi_{\text{graph}}(\cdot)$ and $\Phi_{\text{param}}(\cdot)$ is reduced, which further reduces the computational cost among multiple agents.

#### F.2.2 AGENT'S MODEL INFERENCE

Given the evaluation set $\widetilde{\mathcal{D}}_i^{(t)} = \left\{\widetilde{\mathcal{X}}_i^{(t)},\widetilde{\mathcal{Y}}_i^{(t)}\right\}$ at timestamp $t$, agents calculate the predictions according to their learned model $\boldsymbol{\theta}_i^{(t)}$ and trained backbone $\phi_\beta(\cdot)$ by $f_{\boldsymbol{\theta}_i^{(t)}}\circ\phi_\beta\left(\widetilde{\mathcal{X}}_i^{(t)}\right)$. Here the model parameters are learned from the process $\mathcal{F}_{\gamma^*}(\cdot)$ given fixed parameter $\gamma$.

## G THEORETICAL ANALYSIS

### G.1 ANALYSIS TO THEOREM 1

Our analysis starts from the strong convexity of the loss function $\mathcal{L}\left(f_{\boldsymbol{\theta}_i^{(t)}}\left(\mathbf{X}_i^{(t)}\right),\mathbf{Y}_i^{(t)}\right)$. Since the model $f_{\boldsymbol{\theta}_i^{(t)}}(\cdot)$ is linear and the loss function is standard supervised training loss, the loss function

is convex. Also the condition number of the Hessian $\mathbf{H}\left(\boldsymbol{\alpha}_i^{(t)}\right) = \nabla^2_{\boldsymbol{\alpha}_i^{(t)}} \mathcal{L}\left(f_{\boldsymbol{\theta}_i^{(t)}}\left(\mathcal{X}_i^{(t)}\right), \mathcal{Y}_i^{(t)}\right)$ is bounded, which means the eigenvalues are all strictly large than zero. Based on this observation, we provide an upper-bound of $\left\|\boldsymbol{\theta}_i^{(t)} - \boldsymbol{\theta}_i^{(t)*}\right\|_2$ shown in **Lemma** 1.

**Lemma 1** *Suppose for the $i$th agent at time stamp $t$, given training data $\left(\mathbf{X}_i^{(t)}, \mathbf{Y}_i^{(t)}\right)$, let the linear model is $f_{\boldsymbol{\theta}_i^{(t)}}(\cdot)$ with loss function $\mathcal{L}\left(f_{\boldsymbol{\theta}_i^{(t)}}\left(\mathbf{X}_i^{(t)}\right), \mathbf{Y}_i^{(t)}\right)$. Let $\boldsymbol{\alpha}_i^{(t)}$ be the point doing Taylor expansion and the optimal parameter of the loss function is $\boldsymbol{\theta}_i^{(t)*}$. Then:*

$$\left\|\boldsymbol{\theta}_i^{(t)} - \boldsymbol{\theta}_i^{(t)*}\right\|_2 \leq \frac{L}{2}\left\|\mathbf{H}\left(\boldsymbol{\alpha}_i^{(t)}\right)^{-1}\right\|\left\|\boldsymbol{\alpha}_i^{(t)} - \boldsymbol{\theta}_i^{(t)*}\right\|_2^2,$$

*where $L$ is the Lipschitz constant of the Hessian $\mathbf{H}\left(\boldsymbol{\alpha}_i^{(t)}\right)$.*

**Proof 1** *For the sake of brevity, we write $\boldsymbol{\theta}_i^{(t)}$, $\boldsymbol{\theta}_i^{(t),*}$ and $\boldsymbol{\alpha}_i^{(t)}$ as $\boldsymbol{\theta}$, $\boldsymbol{\theta}^*$ and $\boldsymbol{\alpha}$. Consider the error between $\boldsymbol{\theta}$ and $\boldsymbol{\theta}^*$, we have*

$$\boldsymbol{\theta} - \boldsymbol{\theta}^* = \boldsymbol{\alpha} - \boldsymbol{\theta}^* - \mathbf{H}(\boldsymbol{\alpha})^{-1}\nabla_{\boldsymbol{\alpha}}\mathcal{L}$$
$$= \boldsymbol{\alpha} - \boldsymbol{\theta}^* - \mathbf{H}(\boldsymbol{\alpha})^{-1}\left(\nabla_{\boldsymbol{\alpha}}\mathcal{L} - \nabla_{\boldsymbol{\theta}^*}\mathcal{L}\right)$$

*define $h(t) = \nabla_{\boldsymbol{\theta}^* + t(\boldsymbol{\alpha}-\boldsymbol{\theta}^*)}\mathcal{L}$, thus*

$$\hat{\boldsymbol{\theta}} - \boldsymbol{\theta}^* = \boldsymbol{\alpha} - \boldsymbol{\theta}^* - \mathbf{H}(\boldsymbol{\alpha})^{-1}(h(1) - h(0))$$
$$= \boldsymbol{\alpha} - \boldsymbol{\theta}^* - \mathbf{H}(\boldsymbol{\alpha})^{-1}\int_0^1 h'(s)ds$$
$$= \boldsymbol{\alpha} - \boldsymbol{\theta}^* -$$
$$\mathbf{H}(\boldsymbol{\alpha})^{-1}\int_0^1 \mathbf{H}(\boldsymbol{\theta}^* + s(\boldsymbol{\alpha} - \boldsymbol{\theta}^*))\left(\boldsymbol{\alpha} - \boldsymbol{\theta}^*\right)ds$$
$$= \mathbf{H}(\boldsymbol{\alpha})^{-1}$$
$$\times \int_0^1 \left[\mathbf{H}(\boldsymbol{\alpha}) - \mathbf{H}\left(\boldsymbol{\theta}^* + s(\boldsymbol{\alpha} - \boldsymbol{\theta}^*)\right)\right]\left(\boldsymbol{\alpha} - \boldsymbol{\theta}^*\right)ds$$

*taking norm on both sides, we obtain*

$$\left\|\hat{\boldsymbol{\theta}} - \boldsymbol{\theta}^*\right\|_2 =$$
$$\left\|\mathbf{H}(\boldsymbol{\alpha})^{-1}\int_0^1 \left[\mathbf{H}(\boldsymbol{\alpha}) - \mathbf{H}\left(\boldsymbol{\theta}^* + s(\boldsymbol{\alpha} - \boldsymbol{\theta}^*)\right)\right]\left(\boldsymbol{\alpha} - \boldsymbol{\theta}^*\right)ds\right\|_2$$
$$\leq \left\|\mathbf{H}(\boldsymbol{\alpha})^{-1}\right\|$$
$$\times \left\|\int_0^1 \left[\mathbf{H}(\boldsymbol{\alpha}) - \mathbf{H}\left(\boldsymbol{\theta}^* + s(\boldsymbol{\alpha} - \boldsymbol{\theta}^*)\right)\right]\left(\boldsymbol{\alpha} - \boldsymbol{\theta}^*\right)ds\right\|_2$$
$$\leq \left\|\mathbf{H}(\boldsymbol{\alpha})^{-1}\right\|$$
$$\times \int_0^1 \left\|\mathbf{H}(\boldsymbol{\alpha}) - \mathbf{H}\left(\boldsymbol{\theta}^* + s(\boldsymbol{\alpha} - \boldsymbol{\theta}^*)\right)\right\|\left\|\boldsymbol{\alpha} - \boldsymbol{\theta}^*\right\|_2 ds$$

*For linear regression task, the hessian matrix $\mathbf{H}(\boldsymbol{\theta})$ remains constant, then $\|\mathbf{H}(\boldsymbol{\alpha}) - \mathbf{H}\left(\boldsymbol{\theta}^* + s(\boldsymbol{\alpha} - \boldsymbol{\theta}^*)\right)\| = 0$, which means $\left\|\hat{\boldsymbol{\theta}} - \boldsymbol{\theta}^*\right\|_2 = 0$.*

*For the linear classification tasks, the hessian matrix is Lipschitz continuous, thus there exists a constant $L$ such that*

$$\|\mathbf{H}(\boldsymbol{\theta}_1) - \mathbf{H}\left(\boldsymbol{\theta}_2\right)\| \leq L\|\boldsymbol{\theta}_1 - \boldsymbol{\theta}_2\|$$

*taking this property back into the former inequality, we obtain*

$$\left\| \hat{\boldsymbol{\theta}} - \boldsymbol{\alpha} \right\|_2 \leq L \left\| \mathbf{H}(\boldsymbol{\alpha})^{-1} \right\| \int_0^1 \left\| \boldsymbol{\alpha} - \boldsymbol{\theta}^* \right\|_2^2 (1 - s) ds$$

$$= \frac{L}{2} \left\| \mathbf{H}(\boldsymbol{\alpha})^{-1} \right\| \left\| \boldsymbol{\alpha} - \boldsymbol{\theta}^* \right\|_2^2$$

Our target is minimizing the error upper bound shown in **Lemma** 1. For regression loss functions, the Hessian is constant, making the Lipschitz constant $L$ equal to zero, which means doing expansion at any point $\boldsymbol{\alpha}_i^{(t)}$ is zero error. For other tasks with non-zero Lipschitz constant $L$ such as classification, this goal is equivalent to solving the following constraint optimization problem:

$$\min_{\boldsymbol{\alpha}_i^{(t)}} \mathbb{E}_{\boldsymbol{\theta}_i^{(t)*}} \left( \left\| \mathbf{H}\left( \boldsymbol{\alpha}_i^{(t)} \right)^{-1} \right\| \left\| \boldsymbol{\alpha}_i^{(t)} - \boldsymbol{\theta}_i^{(t)*} \right\|_2^2 \right)$$

$$s.t. \ 1 \leq k \left( \mathbf{H}\left( \boldsymbol{\alpha}_i^{(t)} \right) \right) \leq M, \tag{23}$$

where the prior distribution of the target optimal parameter $\boldsymbol{\theta}_i^{(t)*}$ is $p\left( \boldsymbol{\theta}_i^{(t)*} \right) \sim \mathcal{N}\left( 0, \boldsymbol{\Sigma} \right)$. Due to the difficulty of capturing the spectral radius of the inverse Hessian, considering the eigenvalues $0 < \lambda_1 \leq \lambda_2, \ldots, \leq \lambda_K$ of the objective hessian $\mathbf{H}\left( \boldsymbol{\alpha}_i^{(t)} \right)$, we obtain

$$\frac{K}{\sum_{i=1}^K \lambda_i} \leq \left\| \mathbf{H}\left( \boldsymbol{\alpha}_i^{(t)} \right)^{-1} \right\| = \frac{1}{\lambda_1} \leq \frac{KM}{\sum_{i=1}^K \lambda_i},$$

which means $\left\| \mathbf{H}\left( \boldsymbol{\alpha}_i^{(t)} \right)^{-1} \right\|$ and $\mathbf{tr}\left( \mathbf{H}\left( \boldsymbol{\alpha}_i^{(t)} \right) \right)^{-1}$ are the same order. Hence we obtain the following final optimization problem:

$$\min_{\boldsymbol{\alpha}_i^{(t)}} \mathbb{E}_{\boldsymbol{\theta}_i^{(t)*}} \left( \mathbf{tr}\left( \mathbf{H}\left( \boldsymbol{\alpha}_i^{(t)} \right) \right)^{-1} \left\| \boldsymbol{\theta}_i^{(t)} - \boldsymbol{\theta}_i^{(t)*} \right\|_2^2 \right)$$

$$s.t. \ 1 \leq k \left( \mathbf{H}\left( \boldsymbol{\alpha}_i^{(t)} \right) \right) \leq M. \tag{24}$$

Then we give our proof to **Theorem** 1 in terms of a classification model with $C$ types of output classes.

**Proof 2** *For the sake of brevity, we write $\boldsymbol{\theta}_i^{(t)}, \boldsymbol{\alpha}_i^{(t)}$ as $\boldsymbol{\theta}$ and $\boldsymbol{\alpha}$. The linear model $f_{\boldsymbol{\theta}}(\cdot)$ corresponds to the $i$th class of classification task is $\boldsymbol{\theta}^i$ with training data $\left( \mathbf{X} \in \mathbb{R}^{n \times p}, \mathbf{Y} \in \mathbb{R}^{n \times C} \right)$ where $C$ equals to the number of classes, $p$ is the dimension of input data. The model estimates the probability $\hat{p}^j \in \mathbb{R}^C$ of the $j$th element $\boldsymbol{x}_j$ as*

$$\hat{p}_i^j = \exp\left( \boldsymbol{\theta}^{i\top} \boldsymbol{x}_j \right) / \sum_{i=1}^C \exp\left( \boldsymbol{\theta}^{i\top} \boldsymbol{x}_j \right)$$

*Hence the Hessian equals to*

$$\mathbf{H}\left( \boldsymbol{\alpha} \right) = \sum_{j=1}^n \mathbf{M}_j \otimes \boldsymbol{x}_j \boldsymbol{x}_j^\top,$$

*where $\mathbf{M}_j$ is defined as*

$$\mathbf{M}_j = \begin{pmatrix} \hat{p}_1^j \left( 1 - \hat{p}_1^j \right) & -\hat{p}_1^j \hat{p}_2^j & \cdots & -\hat{p}_1^j \hat{p}_C^j \\ \vdots & \hat{p}_2^j \left( 1 - \hat{p}_2^j \right) & & \vdots \\ -\hat{p}_C^j \hat{p}_1^j & \cdots & \cdots & \hat{p}_C^j \left( 1 - \hat{p}_C^j \right) \end{pmatrix}.$$

*Based on these definitions and notations, the bound shown in **Theorem** 1 can be reformulated as*

$$\boldsymbol{tr}(\mathbf{H}(\boldsymbol{\alpha}))^{-1}\mathbb{E}_{\boldsymbol{\theta}^*}\left(\|\boldsymbol{\alpha}\|_2^2 - 2\boldsymbol{\alpha}^\top\boldsymbol{\theta}^* + \|\boldsymbol{\theta}^*\|_2^2\right)$$

$$= \boldsymbol{tr}(\mathbf{H}(\boldsymbol{\alpha}))^{-1}\left(\|\boldsymbol{\alpha}\|_2^2 - 2\boldsymbol{\alpha}^\top\mathbb{E}_{\boldsymbol{\theta}^*}(\boldsymbol{\theta}^*) + \mathbb{E}_{\boldsymbol{\theta}^*}\left(\|\boldsymbol{\theta}^*\|_2^2\right)\right)$$

$$= \boldsymbol{tr}(\mathbf{H}(\boldsymbol{\alpha}))^{-1}\left(\|\boldsymbol{\alpha}\|_2^2 + \boldsymbol{tr}(\Sigma^{-1})\right)$$

$$\geq \boldsymbol{tr}(\mathbf{H}(\boldsymbol{\alpha}))^{-1}\boldsymbol{tr}(\Sigma^{-1})$$

*However, for $\boldsymbol{tr}\left(\mathbf{H}(\boldsymbol{\alpha})\right)$ we have*

$$\boldsymbol{tr}(\mathbf{H}(\boldsymbol{\alpha})) = \boldsymbol{tr}\left(\sum_{j=1}^n \mathbf{M}_j \otimes \boldsymbol{x}_j\boldsymbol{x}_j^\top\right)$$

$$= \sum_{j=1}^n \boldsymbol{tr}\left(\mathbf{M}_j \otimes \boldsymbol{x}_j\boldsymbol{x}_j^\top\right)$$

$$= \sum_{j=1}^n \|\boldsymbol{x}_j\|_2^2 \sum_{i=1}^C \hat{p}_i^j\left(1 - \hat{p}_i^j\right)$$

$$= \sum_{j=1}^n \|\boldsymbol{x}_j\|_2^2 \left(1 - \sum_{i=1}^C \hat{p}_i^{j2}\right) \leq \left(1 - \frac{1}{C}\right)\sum_{j=1}^n \|\boldsymbol{x}_j\|_2^2$$

*Taking this inequality back into the original bound, we have*

$$\boldsymbol{tr}(\mathbf{H}(\boldsymbol{\alpha}))^{-1}\mathbb{E}_{\boldsymbol{\theta}^*}\left(\|\boldsymbol{\alpha}\|_2^2 - 2\boldsymbol{\alpha}^\top\boldsymbol{\theta}^* + \|\boldsymbol{\theta}^*\|_2^2\right)$$

$$\geq \boldsymbol{tr}(\boldsymbol{\Sigma}^{-1})\left[\left(1 - \frac{1}{C}\right)\sum_{i=1}^n \|\boldsymbol{x}_i\|_2^2\right]^{-1}.$$

*The equality is satisfied when $\hat{p}_1 = \hat{p}_2 = \ldots = \hat{p}_c$ and $\|\boldsymbol{\alpha}\|_2^2 = 0$, which is equivalent to $\boldsymbol{\alpha} = 0$.*

## G.2 ANALYSIS TO THEOREM 2

Recall that the optimal solution corresponds to the first-order condition shown in equation 22. However, this condition requires each agent to know $\mathbf{W}_{ij}^{(t)}$ and $\mathbf{W}_{ji}^{(t)}$ both. In DeLAMA, our real message passing mechanism equation ?? only requires the agent to know $\mathbf{W}_{ij}^{(t)}$ for the $i$th agent. In the following lemma, we first claim that this mechanism does converge to the exact optimal solution by demonstrating that $\mathbf{W}^{(t)}$ is symmetric.

**Lemma 2** *The collaboration graph structure $\mathbf{W}^{(t)} \in \mathbb{R}^{N \times N}$ learned from **Algorithm** 2 is symmetric.*

The proof comes from a simple observation. First rewrite the optimization problem equation 20:

$$\min_{\mathbf{W}^{(t)}} \mathcal{L}\left(\mathbf{W}^{(t)}\right) = \lambda_2\|\mathbf{W}^{(t)} \odot \mathbf{D}^{(t)}\|_1 + \lambda_3\|\mathbf{W}^{(t)}\|_{\mathbf{F}}^2$$

$$\text{s.t. } \|\mathbf{W}^{(t)}\|_1 = m, \ \mathbf{W}^{(t)} \geq 0, \ \text{diag}\left(\mathbf{W}^{(t)}\right) = \mathbf{0}$$

where $\mathbf{D}_{ij}^{(t)}$ stands for $\left\|\boldsymbol{\theta}_i^{(t)} - \boldsymbol{\theta}_j^{(t)}\right\|_2^2$ and $\odot$ is element wise production. Then the observation is, for any feasible solution $\mathbf{W}_0^{(t)}$ of this optimization problem, $\mathbf{W}_0^{(t)\top}$ is also feasible and $\mathcal{L}\left(\mathbf{W}_0^{(t)}\right) = \mathcal{L}\left(\mathbf{W}_0^{(t)\top}\right)$. In fact, we have the relationship $\lambda_2\left\|\mathbf{W}_0^{(t)\top} \odot \mathbf{D}^{(t)}\right\|_1 + \lambda_3\left\|\mathbf{W}_0^{(t)\top}\right\|_{\mathbf{F}}^2 = \lambda_2\left\|\mathbf{W}_0^{(t)\top} \odot \mathbf{D}^{(t)\top}\right\|_1 + \lambda_3\left\|\mathbf{W}_0^{(t)\top}\right\|_{\mathbf{F}}^2 = \lambda_2\left\|\mathbf{W}_0^{(t)} \odot \mathbf{D}^{(t)}\right\|_1 + \lambda_3\left\|\mathbf{W}_0^{(t)}\right\|_{\mathbf{F}}^2$, which means the value of objective function remains unchanged. Thus the proof of **Lemma** 2 is as follows:

**Proof 3** $\|\mathbf{W}^{(t)} \odot \mathbf{D}^{(t)}\|_1$ *is convex, thus $\mathcal{L}\left(\mathbf{W}^{(t)}\right)$ is strongly convex, making the optimization problem with a unique solution $\mathbf{W}^{(t)*}$. However, from our observation, the transpose of $\mathbf{W}^{(t)*}$ is also a feasible solution, and $\mathcal{L}\left(\mathbf{W}^{(t)*}\right) = \mathcal{L}\left(\mathbf{W}^{(t)*\top}\right)$, thus $\mathbf{W}^{(t)*} = \mathbf{W}^{(t)*\top}$, which means the collaboration graph structure learned from the optimization is symmetric.*

Based on **Lemma** 2, the message-passing mechanism does converge to the optimal solution, hence we start our proof to **Theorem** 2.

**Proof 4** *For the sake of brevity, we write $\mathbf{B}_i^{(t)}$, $\boldsymbol{\theta}_i^{(t,k)}$, $\mathbf{W}^{(t)}$ as $\mathbf{B}_i$, $\boldsymbol{\theta}_i$, $\mathbf{W}$. Define*

$$\boldsymbol{\theta}^k = \begin{bmatrix} \boldsymbol{\theta}_1^k \\ \boldsymbol{\theta}_2^k \\ \vdots \\ \boldsymbol{\theta}_N^k \end{bmatrix}, \mathbf{b} = \begin{bmatrix} b_1^k \\ b_2^k \\ \vdots \\ b_N^k \end{bmatrix}, \mathbf{M} = \begin{bmatrix} \mathbf{B}_1^{-1} & 0 & \dots & 0 \\ 0 & \mathbf{B}_2^{-1} & \dots & 0 \\ \vdots & \vdots & \ddots & \vdots \\ 0 & 0 & \dots & \mathbf{B}_N^{-1} \end{bmatrix},$$

*then the iterative algorithm simplifies to:*

$$\boldsymbol{\theta}^k = \mathbf{M} \left( 4\lambda_2 \mathbf{W} \otimes \mathbf{I} \boldsymbol{\theta}^{k-1} + \mathbf{b} \right).$$

*Suppose the optimal parameter is $\boldsymbol{\theta}^*$, then we have*

$$\left\| \boldsymbol{\theta}^k - \boldsymbol{\theta}^* \right\|_2^2 = \left\| 4\lambda_2 \mathbf{M}\mathbf{W} \otimes \mathbf{I} \left[ \boldsymbol{\theta}^{k-1} - \boldsymbol{\theta}^* \right] \right\|_2^2$$

*We write $\|\mathbf{M}\mathbf{W} \otimes \mathbf{I}\|$ as the operator norm of $\mathbf{M}\mathbf{W} \otimes \mathbf{I}$. Then we have*

$$\|4\lambda_2 \mathbf{M}\mathbf{W} \otimes \mathbf{I}\mathbf{x}\|_2^2 = \sum_{i=1}^N \left\| \sum_{j=1}^N 4\lambda_2 \mathbf{W}_{ij} \mathbf{B}_i^{-1} x_j \right\|_2^2$$

$$\leq \sum_{i=1}^N \sum_{j=1}^N \left\| 4\lambda_2 \mathbf{W}_{ij} \mathbf{B}_i^{-1} x_j \right\|_2^2$$

$$\leq \sum_{i=1}^N \sum_{j=1}^N \left\| 4\lambda_2 \mathbf{W}_{ij} \mathbf{B}_i^{-1} \right\|^2 \|x_j\|_2^2$$

$$\leq \sum_{i=1}^N \left[ \sum_{j=1}^N \left\| 4\lambda_2 \mathbf{W}_{ij} \mathbf{B}_i^{-1} \right\| \|x_j\|_2 \right]^2$$

*where $\mathbf{x} \in \mathbb{R}^{Nd} = [x_1^\top; \dots; x_N^\top]^\top$. Define $\tilde{x} = [\|x_1\|; \dots; \|x_N\|]^\top$, $\tilde{\mathbf{M}}_{ij} = \left\| 4\lambda_2 \mathbf{W}_{ij} \mathbf{B}_i^{-1} \right\|$. Then we can conclude that*

$$\|4\lambda_2 \mathbf{M}\mathbf{W} \otimes \mathbf{I}\mathbf{x}\|_2^2 \leq \left\| \tilde{\mathbf{M}} \tilde{x} \right\|_2^2 \leq \left\| \tilde{\mathbf{M}} \right\|^2 \|\tilde{x}\|_2^2 = \left\| \tilde{\mathbf{M}} \right\|^2 \|\mathbf{x}\|_2^2$$

*Recall that the loss function $\mathcal{L}(\boldsymbol{\theta}_i)$ is convex, means we have an eigendecomposition $\mathbf{A}_i = \mathbf{Q}_i^\top \boldsymbol{\Lambda}_i \mathbf{Q}_i$. Taking into the definition of $\mathbf{B}_i$, we have*

$$4\lambda_2 \mathbf{W}_{ij} \mathbf{B}_i^{-1} = 4\lambda_2 \mathbf{W}_{ij} \left[ \mathbf{Q}_i^\top \boldsymbol{\Lambda}_i \mathbf{Q}_i + (2\lambda_1 + 4\lambda_2 d_i) \mathbf{I} \right]^{-1}$$

$$= \mathbf{Q}_i^\top \left[ \frac{\boldsymbol{\Lambda}_i + (2\lambda_1 + 4\lambda_2 d_i) \mathbf{I}}{4\lambda_2 \mathbf{W}_{ij}} \right]^{-1} \mathbf{Q}_i$$

*Thus the operator norm of $4\lambda_2 \mathbf{W}_{ij} \mathbf{B}_i^{-1}$ equals to*

$$\left\| 4\lambda_2 \mathbf{W}_{ij} \mathbf{B}_i^{-1} \right\| = \frac{4\lambda_2 \mathbf{W}_{ij}}{\lambda_i^* + (2\lambda_1 + 4\lambda_2 d_i)},$$

*where $\lambda_i^* > 0$ is the minimum eigenvalue of $\boldsymbol{\Lambda}_i$.*

*By Gershgorin circle theorem, the eigenvalues of $\tilde{\mathbf{M}}$ is bounded by*

$$\lambda < \sum_{j=1}^N \tilde{\mathbf{M}}_{ij}$$

$$= \sum_{j=1}^N \frac{4\lambda_2 \mathbf{W}_{ij}}{\lambda_i^* + (2\lambda_1 + 4\lambda_2 d_i)} = \frac{4\lambda_2 d_i}{\lambda_i^* + (2\lambda_1 + 4\lambda_2 d_i)},$$

*for some $1 \leq i \leq N$. Thus*

$$\left\| \tilde{\mathbf{M}} \right\| < \max_{1 \leq i \leq N} \left[ \frac{4\lambda_2 d_i}{2\lambda_1 + 4\lambda_2 d_i} \right] = \rho < 1,$$

*apply this property to $\boldsymbol{\theta}^{k-1}$ we have*

$$\left\| \boldsymbol{\theta}^k - \boldsymbol{\theta}^* \right\|_2^2 < \rho \left\| \boldsymbol{\theta}^{k-1} - \boldsymbol{\theta}^* \right\|_2^2.$$

*Taking $k$ goes to infinity we have $\left\| \boldsymbol{\theta}^k - \boldsymbol{\theta}^* \right\|_2^2$ goes to zero with a linear convergence rate.*

## H    EXPERIMENTAL DETAILS

### H.1    REGRESSION PROBLEM

#### H.1.1    EXPERIMENTAL SETUP

**Task.** Following the cognitive scientific principles detailed in Almaatouq et al. (2020), we design a system with 6 agents in this collaborative regression setting, where the collaboration relationship is easy to verify. We design several non-linear functions to generate data points and each agent accesses a small dataset at each time drawn from one of the non-linear functions. One function can be accessed by multiple agents. Each agent is required to regress the whole corresponding function. Since the data points of one single agent are not informative enough, correctly collaborating with other agents corresponding to the same function is essential. Moreover, to increase the difficulty of the collaborative relationship inferring, agents are categorized into three types based on data quality: i) type one, agents possess numerous samples with minimal noise; ii) type two, agents have a limited sampling range with medium noise; and iii) type three, agents access very few samples with significant noise. This agent categorization requires the agent to consider both the function correspondence and data quality during collaboration.

**Dataset.** The total function number ranges randomly from 1 to 3 and each function is ensured to correspond to at least two agents. The data points are obtained by sampling points from a function $f(x)$ by $\hat{y} = f(x) + \epsilon$, with a Gaussian noise $\epsilon \sim \mathcal{N}(0, \sigma)$. The functions are random linear combinations of quadratic functions and sinusoidal functions, with domain [-5, 5]. At each time, different types of agents' training sets have different sample ranges: type one agents' training set is generated from the full domain [-5, 5], while type two and type three agents' training sets are generated from a random limited interval with length 1. We generated 600 task learning sequences with different ground truth functions, with 2/3 used for training, and 1/3 used for testing.

**Evaluation metric.** We adopt two metrics for evaluation: i) the average mean square error between the agent regression points and the ground-truth function's points at each time stamp $t$; ii) the graph mean square error, which is defined as the normalized distance between the learned collaboration graph and the oracle collaboration graph at time $t$:

$$\text{GMSE} = \frac{\left\| \mathbf{W}^{(t)} - \mathbf{W}_{\text{oracle}} \right\|_{\mathbf{F}}}{\left\| \mathbf{W}_{\text{oracle}} \right\|_{\mathbf{F}}}. \tag{25}$$

The oracle graph adjacency matrix $\mathbf{W}_{\text{oracle}}$ is defined as $\mathbf{W}_{\text{oracle}} = m \frac{\widetilde{\mathbf{W}}}{\|\widetilde{\mathbf{W}}\|_1}$, with $\widetilde{\mathbf{W}}_{ij} = 1$ if agent $i$ and $j$ share a same function and $\widetilde{\mathbf{W}}_{ij} = 0$ otherwise. Note that the oracle graph represents a relatively optimal solution for collaboration since it acquires the agent-function correspondence but may not be the absolute optimal solution since the agent data quality is not considered.

#### H.1.2    QUANTITATIVE ANALYSIS

Here we aim to: i) verify the effectiveness of the decentralized collaboration mechanism in `DeLAMA` on regression learning performance; ii) evaluate the collaboration graph learning performance of `DeLAMA` under different constraints of the communication graph $\mathbf{C}^{(t)}$. Note that in i) the communication structure of `DeLAMA` is set to be fully connected, while in ii) we will try other kinds of communication structures.

Table 4: Performance comparison for the regression problem. We compare our method with representative federated learning approaches and decentralized optimization methods. We also perform an ablation study of our approach, including `DeLAMA-WC` and `DeLAMA-WM`. The former stands for `DeLAMA` without collaboration among agents while the latter stands for `DeLAMA` without lifelong-adaptive learning capabilities.

| Setting | Method | $\text{MSE}_{t=1}$ | $\text{MSE}_{t=10}$ |
|---|---|---|---|
| **Federated learning** | FedAvg McMahan et al. (2017) | 12.9147 | 10.3229 |
| | FedProx Li et al. (2020) | 11.8117 | 10.5989 |
| | SCAFFOLD Karimireddy et al. (2020) | 13.6683 | 12.8070 |
| | FedAvg-FT McMahan et al. (2017) | 9.8845 | 6.7536 |
| | FedProx-FT Li et al. (2020) | 9.4131 | 6.6035 |
| | Ditto Li et al. (2021) | 16.3461 | 7.9752 |
| | FedRep Collins et al. (2021) | 40.0225 | 11.4844 |
| | pFedGraph Ye et al. (2023) | 8.6481 | 5.9771 |
| **Decentralized optimization** | DSGD Lian et al. (2017) | 23.8660 | 12.6872 |
| | DSGT Pu and Nedić (2018) | 17.7407 | 17.7413 |
| | DFedAvgM Sun et al. (2022) | 13.2054 | 10.4894 |
| **Decentralized and lifelong-adaptive collaborative learning** | `DeLAMA-WC` | 5.0794 | 0.0785 |
| | `DeLAMA-WM` | **2.3377** | 2.6150 |
| | `DeLAMA` | **2.3377** | **0.0719** |

Table 5: The evaluation of the collaboration performance compared with other centralized collaboration mechanisms. `DeLAMA` effectively learns collaboration relationships under different communication structure constraints.

| $\mathcal{G}^{(t)}$ | $c(\mathcal{G}^{(t)})$ | Method | Decentralize | $\text{GMSE}_{t=1}$ | $\text{GMSE}_{t=5}$ | $\text{GMSE}_{t=10}$ | $\text{MSE}_{t=1}$ | $\text{MSE}_{t=5}$ | $\text{MSE}_{t=10}$ |
|---|---|---|---|---|---|---|---|---|---|
| **FC** | 1 | NS | ✓ | 1.4198 | 1.4900 | 1.4396 | 56.1900 | 0.8384 | 0.3919 |
| | | GLasso | ✗ | 0.9687 | 0.9449 | 0.9540 | 3.9664 | 0.1923 | 0.0817 |
| | | MTRL | ✗ | 72.26 | 3.955 | 60.3069 | 2.6570 | 0.3009 | 0.1837 |
| | | GL-LogDet | ✗ | 1.0391 | 0.8916 | 0.8331 | 3.2395 | 0.1416 | 0.0726 |
| | | L2G-PDS | ✗ | 3.6415 | 1.2675 | 0.6638 | 4.4638 | 0.1633 | 0.0744 |
| | | Oracle | - | - | - | - | 1.1611 | 0.0847 | 0.0695 |
| | | `DeLAMA` | ✓ | **0.2992** | **0.0930** | **0.0721** | **2.3377** | **0.1104** | **0.0719** |
| **ER** | 0.3 | `DeLAMA` | ✓ | 1.8291 | 1.7683 | 1.7712 | 3.0061 | 0.2176 | 0.0944 |
| | 0.5 | | | 1.1627 | 0.9794 | 0.9736 | 2.8659 | 0.1654 | 0.0850 |
| **BA** | 0.3 | | | 1.8244 | 1.7697 | 1.7728 | 2.8893 | 0.2391 | 0.0905 |
| | 0.5 | | | 0.9755 | 0.8475 | 0.8371 | 2.7184 | 0.1781 | 0.0787 |

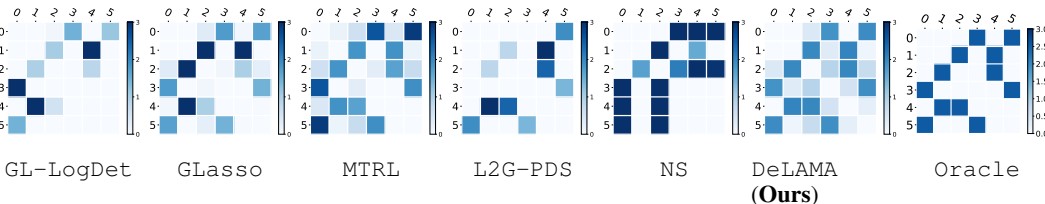

|     GL-LogDet     |     GLasso     |     MTRL     |     L2G-PDS     |     NS     |     DeLAMA (**Ours**)     |     Oracle     |

Figure 5: Visualization of the learned graph structure at time $t = 3$ compared with the oracle collaboration structure. `DeLAMA` learns smooth and accurate edge weights compared to baseline methods.

To verify i), **Table** 4 presents the system performance comparison with federated learning and decentralized optimization methods, including standard federated learning McMahan et al. (2017); Li et al. (2020); Karimireddy et al. (2020), personalized federated learning Li et al. (2021); Ye et al. (2023) and Lian et al. (2017); Pu and Nedić (2018); Sun et al. (2022). The experimental result shows that i) our decentralized and lifelong collaboration approach significantly outperforms both federated learning and decentralized optimization methods, reducing the MSE by **98.80**% at last timestamp $t = 10$; ii) adding our agent collaboration remarkably improves the learning performance, especially when timestamp is small that every agent lacks data; iii) as timestamp increases, the learning performance improvement brought by our lifelong-adaptive learning enlarges, reflecting the effectiveness of our lifelong-adaptive learning design.

To verify ii), we perform `DeLAMA` under different communication structures, including the fully connected(FC) graph, the Erdos-Renyi(ER) graph, and the Barabasi Albert(BA) graph Barabási (2013). The compared approaches contain both classic relation learning methods Banerjee et al. (2008); Liu et al. (2017), and recent widely used structural learning approaches L2G Pu et al. (2021). We compare these approaches by substituting $\mathbf{\Phi}_{\mathrm{graph}}(\cdot)$ of `DeLAMA` to other relational learning methods while preserving other parts of `DeLAMA` stay unchanged. To achieve a fair comparison, we investigate the system performance under the same connectivity level of these two types of random graphs. Here the connectivity level $c\left(\mathcal{G}^{(t)}\right)$ of the communication graph adjacency matrix $\mathbf{C}^{(t)} \in \{0,1\}^{N \times N}$ is defined as: $c\left(\mathcal{G}^{(t)}\right) = \|\mathbf{C}^{(t)}\|_1 / N^2$. **Table** 5 presents the comparison with previous relational learning methods on both the structural correctness and the task performance. The results show that i) `DeLAMA` outperforms both traditional centralized and decentralized structural learning methods; ii) `DeLAMA` matches the performance of recent structural learning performance L2G Pu et al. (2021); and iii) `DeLAMA` stills works even under low connectivity level.

### H.1.3 QUALITATIVE ANALYSIS

Here we aim to i) verify that the system of `DeLAMA` could dynamically evolve as time increases, and ii) compare the performances of our collaboration relationship inference with existing graph learning approaches, including NS Meinshausen and Bühlmann (2006), GLasso Banerjee et al. (2008), MTRL Liu et al. (2017), GL-LogDet Dong et al. (2016) and L2G Pu et al. (2021).

To show i), we visualize the collaborative learning results of each agent from time stamp 1 to 5, and the corresponding collaborative relationships. **Figure** 6 illustrates the collaboration learning results of two groups among six agents. The visualizations show that agents can find correct collaborators where the connection forms two groups of agents, and refine the time-evolving collaboration structures as time increases. The visualization of their learned functions shows that they can learn from past encountered data samples.

To show ii), we visualize the system collaboration relationship learned by different approaches. To achieve a fair comparison, we calculate the collaboration relationship for the group of agents at the same time $t = 3$. The visualization of collaboration relationships shown in **Figure** 5 demonstrates that `DeLAMA` could discover more suitable collaboration relations compared to other approaches.

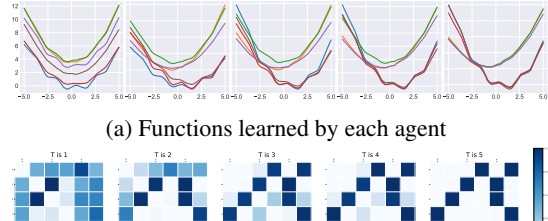

(a) Functions learned by each agent

(b) Time evolving collaboration structure

Figure 6: A collaboration result of two groups of agents that are performing two different regression tasks. (a) The learned functions after collaboration for each agent. (b) The time-evolving collaboration graph of this collaboration system. `DeLAMA` enables the collaboration system to evolve with dynamic collaboration relationships.

### H.2 REGRESSION PROBLEM

**Implementation.** We use a fully connected network as the backbone to represent the nonlinear transform $\phi$ as part of the training parameters of `DeLAMA`. The number of hidden layers is 2 with the output layer set in dimension 50. The batch size used for training is 2 with a learning rate of

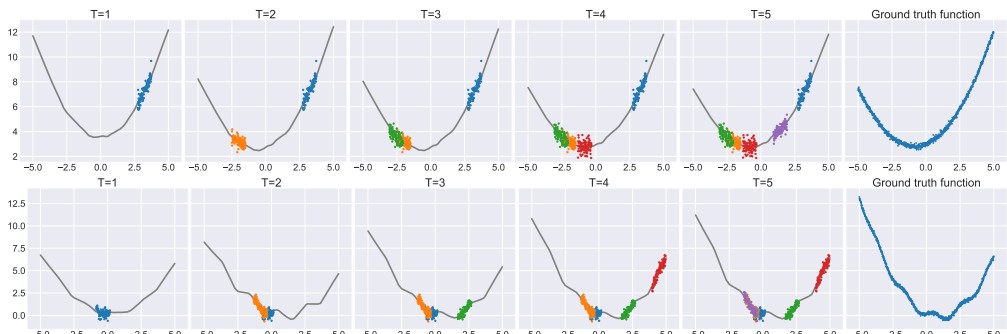

Figure 7: A time-evolving learning task example of two agents sampled from the regression task dataset. For each agent, the training samples are generated around the agent's ground truth function with a limited sampling interval. The agent's knowledge of the target function will improve as the number of sampling intervals increases.

1e-3. The iteration numbers for collaborative relational inference $\Phi_{graph}$ and lifelong model update $\Phi_{param}$ are set to $M_1 = 10$ and $M_2 = 10$, respectively.

**Task configuration.** We visualize one example of the generated task sequences shown in **Figure** 7. In this example, we can see two different agents with distinct learning goals (the black line). From $T = 1$ to $T = 5$, the agents encountered different batches of data samples, forming different viewpoints of the target regression function. In the beginning, the agents could not have a full understanding of their learning task. As time increases, agents could be able to guess the learning target function with the help of their lifelong learning ability.

### H.3 IMAGE CLASSIFICATION

**Implementation.** We used self-designed convolution network in **MNIST** tasks and ResNet-18 in **CIFAR-10** tasks to represent the backbone for nonlinear transform $\phi_\beta(\cdot)$. For the self-designed convolution network, the convolution layer number is set to 2 with the output feature dimension 50. For the **CIFAR-10** tasks we removed the batch normalization layers of ResNet-18 to increase the training stability, and the output feature dimension is also set to 50. The batch size is 2 with a learning rate of 1e-3. The iteration number for lifelong model update $\Phi_{param}(\cdot)$ is set to 10 and the number of iteration steps of collaboration relational inference $\Phi_{graph}(\cdot)$ is set to 5. For federated learning methods, we run 100 rounds for each timestamp. During each round, each client conducts 5 iterations of model training for **MNIST** and **CIFAR-10**.

### H.4 MULTI-ROBOT MAPPING

**Implementation.** The backbone used for `DeLAMA` is defined as a simple multi-layer perceptron with 2 hidden layers. The output feature dimension of the MLP is 10 and the local models' output dimension is 1 followed by a sigmoid activation function to represent the occupancy probability. For the training of the unrolled network part, the learning rate is 1e-3 with batch size 2. The iteration number of message passing is set to 10 and the graph learning iteration number is set to 5.

### H.5 HUMAN INVOLVED EXPERIMENT

### H.5.1 EXPERIMENTAL SETUP

**Purposes.** Here we designed a human experiment for multi-agent collaboration. It aims to verify whether human tends to collaborate with other people with similar knowledge, which corresponds to the graph smoothness modeling shown in equation 15. We created a cognitive learning task formed by a collaborative learning game to investigate the following two problems:

• Whether the proposed algorithm obtains a compatible task performance compared to humans in cognitive learning scenarios.

Table 6: The average performance of human collaboration compared with DeLAMA. We analyzed the system performance at $t = 1, 2, 3$.

| Method | Quality | $\text{MSE}_{t=1}$ | $\text{MSE}_{t=2}$ | $\text{MSE}_{t=3}$ |
|---|---|---|---|---|
| Human | Low | 268.9874 | 8.4663 | 0.7053 |
| | Medium | **20.8548** | 1.2173 | 0.0859 |
| | High | 0.0503 | 0.0183 | 0.0270 |
| DeLAMA | Low | **246.9219** | **4.1179** | **0.6533** |
| | Medium | 22.8994 | **0.6001** | **0.0600** |
| | High | **0.0320** | **0.0120** | **0.0112** |

7.5cm

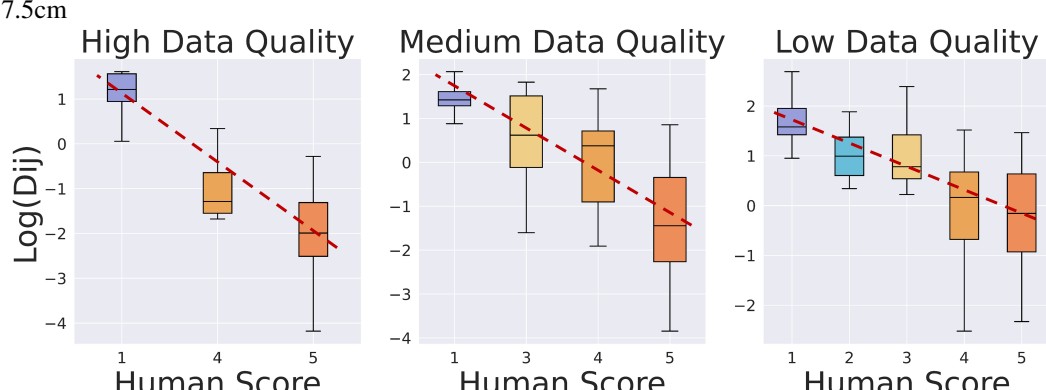

Figure 8: The human evaluation of collaboration relationships with each other. **x-axis:** human rating scores, ranging from 0 (lowest collaboration) to 5 (highest collaboration). **y-axis:** logarithm of model parameter distance. **From the first to last column:** different data quality human evaluation score distribution (highest to lowest). There exist negative correlations between agents' model parameters and human collaboration strength.

• Does human collaboration share the same mechanism to find collaborators like the proposed graph learning algorithm?

**Task.** To demonstrate that DeLAMA's collaboration mechanism mirrors human cognitive collaboration, we adopt a linear regression task as **Section H.1** where each human participant represents one system agent. Here, to promote efficient human-machine interaction, we create a web application with a user-friendly GUI. From the human's perspective, i) the linear regression procedure is performed based on their visual observation and memory of previously encountered data points; ii) the interaction between the human participant and other agents occurs on the visual level, where the human can view the agents' regression outcomes and refine their regression results by drawing on the GUI based on the evaluation of other agents' results; see details in **Appendix H.6**.

### H.5.2 QUANTITATIVE ANALYSIS

To answer the first question, we aim to analyze the average performance of humans compared with DeLAMA. We set the number of agents $n = 4$ with 3 timestamps in total. We set the two different target regression lines between the agents, which corresponds to two possible collaboration groups. We conducted three different levels of data quality. The qualitative results shown in **Table 6** reveal that DeLAMA outperforms human collaboration with a lower regression error at various data qualities.

### H.5.3 QUALITATIVE ANALYSIS

To answer the second question, we aim to compare the correlations between the collaboration relationships and the corresponding agents' differences. From the definition of collaboration relation defined in DeLAMA equation **??**, the collaboration strength is negatively correlated with the distance of model parameters. Here we aim to verify that human collaboration still satisfies this property.

Specifically, we substitute one agent in `DeLAMA` into a human and compare the relationship between learned edge weights and parameter distances. To quantify the collaboration strength between humans and collaborated machines, we ask humans to provide their rating scores (0 to 5, a higher score means a higher collaboration relationship) against other agents. **Figure** 8 demonstrates the relation of human rating scores and learned regression model parameter distances under different task data quality, where the y-axis is the logarithm of the distance defined as $\log\left(\mathbf{D}_{ij}^{(t)}\right) = \log\left(\left\|\boldsymbol{\theta}_i^{(t)} - \boldsymbol{\theta}_j^{(t)}\right\|_2\right)$.
From the human score distribution, we see that 1) the collaboration relation strength is negatively correlated with the distance between the agents' model parameters, which verifies our assumption that humans tend to collaborate with similar ones; and 2) for harder collaboration problems with lower data quality, humans have difficulty in distinguishing useful collaborators. This suggests that the mechanisms of human collaboration are not merely based on the similarity between individuals. There may exist some more complex logic and structures, such as complex gaming relationships and higher-order interactions.

### H.6 HUMAN INVOLVED EXPERIMENT

**System overview.** The collaboration system employs a hierarchical structure. At the user interface layer, it comprises real human users and 'virtual agents' used for human-machine interaction. At the back end are the agents powered by `DeLAMA`. Note that to enable interaction between humans and machines in the back end, we utilize the 'proxy machines' to mimic human behaviors. The full system framework is shown in **Figure** 9.

**Web GUI.** Here we provide a web GUI example to describe how our human-involved experiment works. There are 5 main pages of the web GUI, including the login page (**Figure** 10), the local learning page (**Figure** 11), the result modification page (**Figure** 12), and the output page (**Figure** 13). The human-involved experiment's routine is as follows:

1. **Login.** Each participant will read the user guide at the login page (shown in **Figure** 10). Then input their user name to represent their ID and enter the game.

2. **Local learning.** The participant will first encounter a plot of scatter points (as shown in **Figure** 11), then draw a line to represent their guess of the regression line. This line represents their learning knowledge purely based on their visual information.

3. **Aggregate and collaborate.** The agent will have a view of other participants' learning lines, and guess whether to modify their initial learned regression results according to others' results. Then the submitted final learning result will be evaluated and return the regression loss to the participant as shown in **Figure** 12.

On the final page, the participant will see all their historical learning results and the exact ground truth regression target function as shown in **Figure** 13.

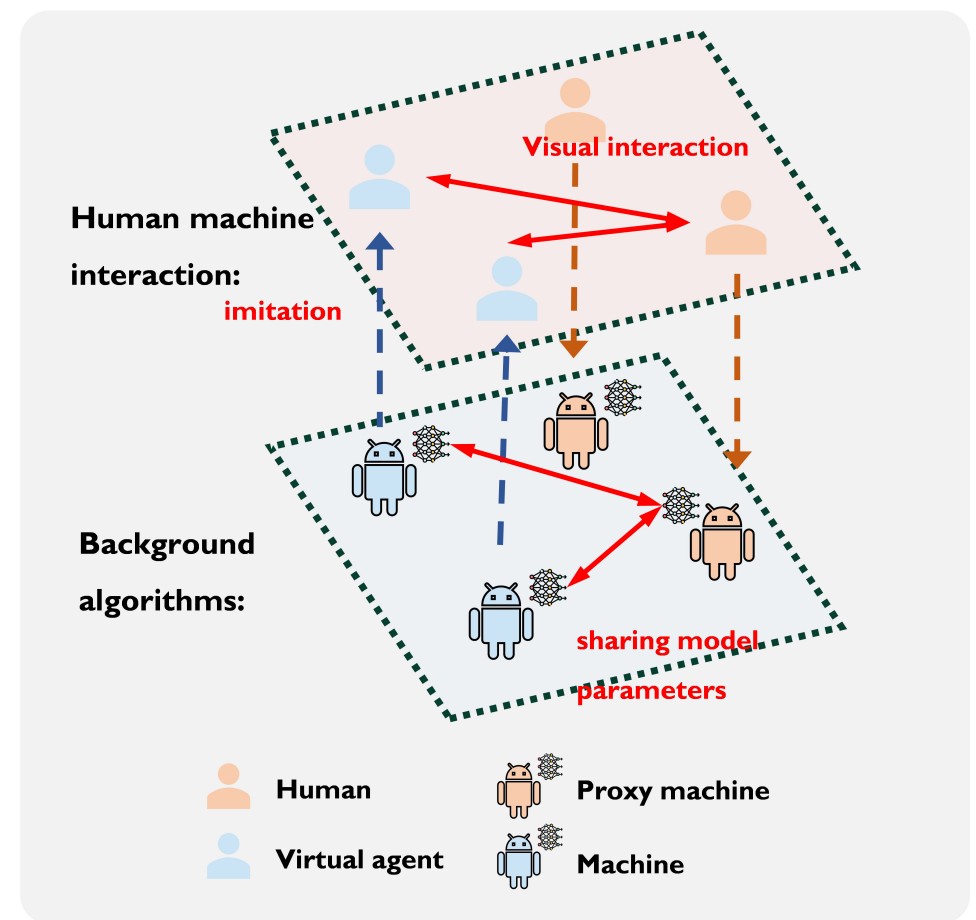

Figure 9: The human-machine interaction framework used for the human-involved experiment. In the interaction phase, agents can observe others' behaviors and submit their understandings. To enable the interaction between humans and agents running in `DeLAMA`, we add virtual agents to perform visual interaction with each other. In the back end is the collaboration algorithm powered by `DeLAMA`, with proxy agents imitating human behaviors. This top-down framework enables human-machine communication and provides a platform for real human evaluation.

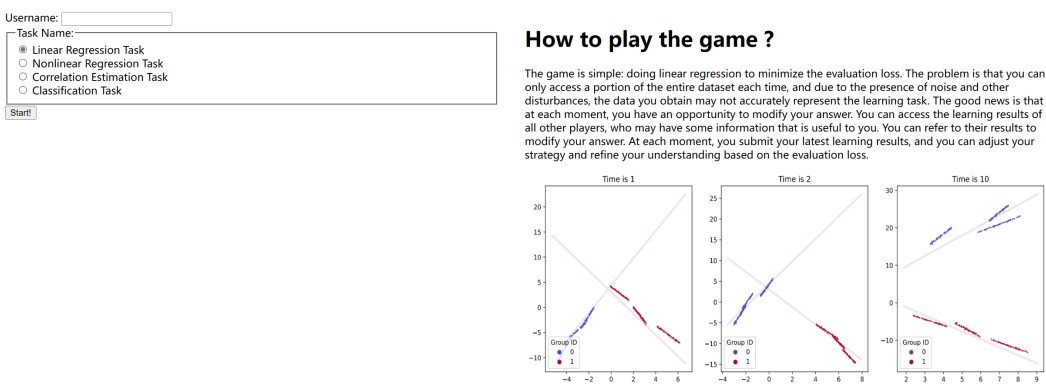

Figure 10: The login page of the human-involved experiment, contains a user guide for the participants to learn how to play the game.

# Local learning

User ID: 0

Username:

Current time: 0

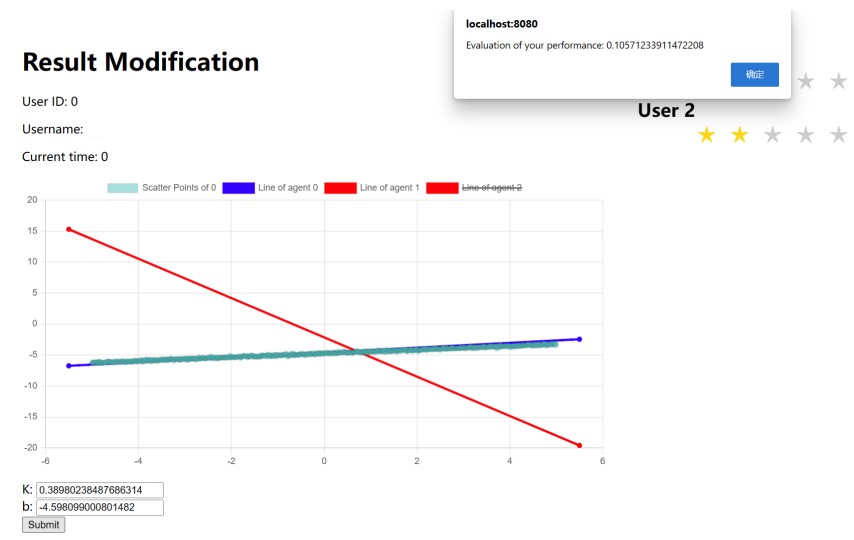

Equation: y = 0.39x + -4.60

K: 0.38980238487686314

b: -4.598099000801482

Submit

Figure 11: The local learning page of the human-involved experiment. Here participant's drawn line will be transformed into a linear equation with two parameters.

## Result Modification

User ID: 0

Username:

Current time: 0

K: 0.38980238487686314

b: -4.598099000801482

Submit

localhost:8080

Evaluation of your performance: 0.10571233911472208

确定

User 2

Figure 12: The aggregation page of the human-involved experiment, which contains other participants' results and the score rated by each agent.

**Game finished**

- Try Again!

**Leaderboard**

| Rank | Name | Avg Performance |
|------|------|-----------------|
| 0 | | 0.039735423565615414 |
| 1 | | 127.4480893376305 |
| 2 | | 187.79727256445176 |

**Result Log**

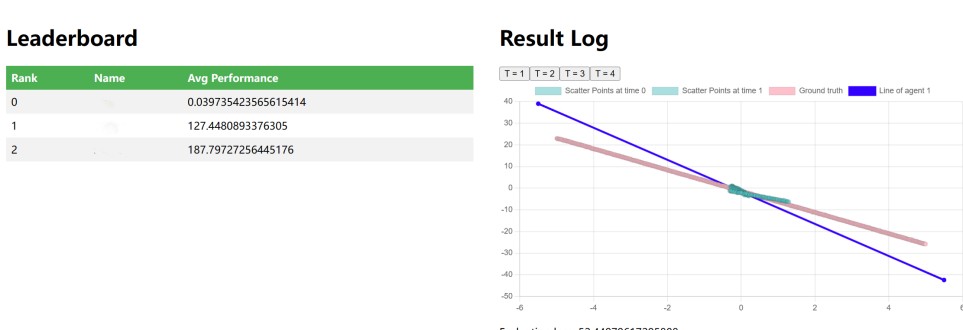

Evaluation loss: 52.44979617295808

Figure 13: The final page of the human-involved experiment with a leaderboard, each participant can look back to historical learning results.

