# OpenReview forum: "Decentralized and Lifelong-Adaptive Multi-Agent Collaborative Learning"
_ICLR.cc/2026/Conference — Submitted to ICLR 2026_

### Official Review · Reviewer_iUuu · 2025-10-28

**Soundness:** 3
**Presentation:** 4
**Contribution:** 2
**Rating:** 6
**Confidence:** 4

**Summary:**

Overall, this paper presents a good and comprehensive contribution to multi-agent learning. It tackles a timely problem, enabling multi-agent to learn collaboratively in a decentralized, continually evolving environment, and proposes a solution that is both theoretically grounded and empirically effective. The optimization formulation is built on sound principles and addresses critical aspects like decentralization, adaptability, and communication efficiency. I found the formulation to be well-motivated and technically sound: by integrating graph structure learning and lifelong learning into a unified problem, the authors capture the core challenges of dynamic multi-agent learning in a principled way. The use of alternating convex optimization for this biconvex problem is appropriate and supported by existing theory, and the authors provide convergence analysis to back up the solver’s reliability.

**Strengths:**

S1) Timely problem & effective solution. The paper formulates decentralized, lifelong-adaptive multi-agent collaboration under communication constraints as Optimization Problem (1) and instantiates a practical solver validated by quantitative and qualitative experiments;

S2) Novel use of algorithm unrolling (clarity & efficacy). The alternating solver is unrolled into a neural mapping with three clear modules , accompanied by design choices that reduce iterations and make hyper-parameters learnable;

S3) Theoretical and empirical support. The paper provides non-trivial analysis and proofs and corroborates them with competitive comparisons and ablations and qualitative evolution;

S4) Ablation studies and component analysis. The ablation with DeLAMA-WC (no collaboration) and DeLAMA-WM (no lifelong adaptation) isolates module contributions and demonstrates the full model’s benefits on regression metrics.

**Weaknesses:**

* **W1) Scalability & Efficiency**

  * Cost of maintaining the global collaboration matrix. The method maintains and updates a full collaboration matrix $W^{(t)} \in \mathbb{R}^{N \times N}$ and performs per-iteration neighbor exchanges, which can become heavy as agents $N$ grow. This matters for scalability and network overhead.
  * Communication bottleneck caused by global aggregation. The graph-inference step includes global gatherings (“Gather $(x^k_j, y^k_j)$ via $G^{(t)}$ to Agent $i$”), implying system-wide reductions/broadcasts that may bottleneck large deployments. This matters for communication complexity.
  * Communication and time overhead of inner loops. Each time step unrolls two inner loops — graph updates ($M_1$) and parameter updates ($M_2$) — before progressing, increasing per-timestamp communication rounds; no explicit practical bounds for $M_1, M_2$ are reported. This matters for latency and throughput.
  * Computational burden of high-dimensional Hessian calculation. The local module explicitly computes Hessians $H_i^{(t)}$ to build $A_i^{(t)}, b_i^{(t)}$ (Algorithm 4), which may be expensive for high-dimensional models. This matters for per-agent compute cost.
  * Lack of runtime and bandwidth measurements. The paper does not provide wall-clock/runtime or bandwidth measurements for the unrolled pipeline or the meta-training stage. This limits empirical evidence on efficiency. No direct evidence found in the manuscript.


* **W2) The Number of Agents**

  * Limited scale of current experiments. The method has so far been validated on only 5–6 agents and has not been tested under larger-scale or multi-task settings. Therefore, its scalability in high-dimensional, large multi-agent systems remains uncertain.

  * Communication cost of the decentralized Newton step. In the worst case, the decentralized Newton method requires multiple rounds of communication among all neighboring agents to complete one update. The communication rounds may increase rapidly with the number of agents, yet the paper does not report the specific communication or time cost of this step.

  * Lack of runtime and complexity analysis. The paper does not provide measured runtime, bandwidth usage, or other empirical statistics, nor does it analyze algorithmic complexity, making it difficult to assess its efficiency and feasibility in larger systems.

  * Suggested experiments for scalability verification. It is recommended that the authors conduct or include additional experiments or simulations to analyze scalability — for example, simulating 20 or more agents, constructing random communication graphs, and reporting how runtime or communication rounds scale with network size.

  * Alternative analytical evidence if large-scale experiments are infeasible. If large-scale experiments are infeasible, the authors could provide theoretical complexity analyses instead, explaining how each agent’s computational and communication costs per iteration grow with the number of neighbors, thereby helping readers evaluate its applicability in large-scale distributed scenarios.

* **W3) Recency of Baselines and Related Comparisons**

  * Recency of baselines. Most of the baselines (And the Related works) in the paper date before 2023 (e.g., those listed in Table 1 and Table 2). If space permits, it is recommended to include or discuss more recent (2024–2025) decentralized or continual collaboration works to better highlight the progress and positioning of this method relative to the latest research.

  * Citation clarity in Tables 1&2. Some algorithm names in Table 1 lack explicit citation markers, making it difficult for readers to identify their corresponding source papers. The authors are advised to add clear citation references for each baseline method within the table or its caption to enhance traceability and academic rigor.

**Questions:**

See the Weaknesses.

**Details Of Ethics Concerns:**

No Details Of Ethics Concerns.

---

### Official Review · Reviewer_H8aM · 2025-10-28

**Soundness:** 3
**Presentation:** 3
**Contribution:** 2
**Rating:** 4
**Confidence:** 3

**Summary:**

The paper “develops DeLAMA for decentralized multiagent learning that enables multiple agents to learn collaboratively in a fully decentralized in a continually adaptive setting without relying on a central server. DeLAMA enables agents to identify useful collaborators and adapt to learn tasks faster. DeLAMA introduces two main components: a decentralized graph structure learning algorithm, which enables agents infer their collaboration strengths based on task similarity, and a lifelong learning mechanism that integrates a compact memory unit to retain past knowledge to mitigate forgetting and to enable continuous adaptation to new data.  DeLAMA also employs algorithm unrolling, transforming its iterative optimization into a neural network that combines the interpretability of optimization theory with the expressive power of deep learning, allowing agents to “learn how to collaborate” through supervised training. Theoretical guarantees demonstrate fast convergence with communication efficiency. Experiments on regression, classification, and multi-robot mapping tasks are provided to demonstrate that DeLAMA outperforms federated and decentralized baselines.

**Strengths:**

1. The paper is well-organized and well-written and can be followed straightforwardly.

2. The idea of enabling autonomously learning collaboration structures and adapting continuously to dynamic tasks is interesting, novel, and of practical importance.

3.Theoretical results are sound and demonstrate convergence guarantee which is very important in decentralized learning settings.

4. Experiments are extensive beyond just demonstrating that DeLAMA works. Various informative analytic and ablative experiments are performed and reported.

5. Evaluation includes regression, classification, and robotics which demonstrate that the proposed method is applicable to various applications.

**Weaknesses:**

1. The optimization involves alternating convex searches, Taylor approximations, and graph Laplacian regularization, then wraps them in algorithm unrolling. This pipeline is computationally heavy and might not be applicable for a lifelong learning setting where learning speed is important. This is different from convergence because we may not many iterations but each each iteration can be very time-consuming.

2. Although theoretically efficient, the Newton-based graph inference and Jacobi-style parameter updates still require multiple message exchanges per iteration. This may create non-trivial latency which can be problematic for lifelong learning settings.

3. Despite experiments on various tasks, the datasets that are used are simple by current standards. It raises the question whether the proposed approach can scale well with more complex tasks that require using complex and large models.

4. Experiments include only 6 agents at most which raises the question of scalability. Experiments need to be expanded and include more agents with more heterogeneity to demonstrate practicality.

5. Baselines that are used for comparison do not include baselines developed in the past three years. It is important to include all recent developments to demonstrate competitive performance.

6. The code is not provided which makes judgment about performance challenging.

**Questions:**

1. How does the computational and communication cost scale with the number of agents? This is a very important aspect that deserves to be studied.

2. The method enforces sparse collaboration via an L1 penalty. How sensitive is performance with respect to the sparsity level?

3. Is it possible to quantify the actual communication load (e.g., data volume, bandwidth, or latency) compared to baselines, e.g., DSGD or FedAvg through experiments?

4. How sensitive is DeLAMA’s performance to the number of unrolled iterations or hyperparameters? Does the user need to do hyperparameter tuning for an optimal performance.

---

### Official Review · Reviewer_EBsY · 2025-10-31

**Soundness:** 2
**Presentation:** 3
**Contribution:** 2
**Rating:** 4
**Confidence:** 3

**Summary:**

The paper presents DeLAMA, a decentralized multi-agent lifelong learning framework that enables agents to learn task-specific models, dynamically infer collaboration relationships, and maintain bounded memory. It combines decentralized graph learning, memory units, and algorithm unrolling to enhance efficiency and adaptability. Experiments show DeLAMA outperforms existing methods in tasks like image classification and multi-robot mapping, with theoretical analysis validating its collaboration mechanism.

**Strengths:**

1.DeLAMA is supported by comprehensive theoretical proofs, ensuring its stability, efficiency, and convergence.
2.Uniquely combines decentralized learning, lifelong learning, and dynamic graph learning into a single framework. This innovative approach eliminates the need for a central server and adapts to real-time task changes, offering a versatile solution for multi-agent systems.
3.The framework enables agents to autonomously adjust their collaboration relationships based on evolving tasks.
4. incorporates algorithm unrolling reduce communication and computational costs while maintaining high expressiveness.

**Weaknesses:**

1.The experimental results do not demonstrate significant or optimal performance improvements compared to existing methods, raising questions about the effectiveness of DeLAMA in achieving superior outcomes.
2.The baseline methods used for comparison are relatively old and not state-of-the-art, which may not provide a fair or comprehensive evaluation of DeLAMA`s capabilities.
3.The experiments are conducted on smaller datasets (e.g., MNIST, CIFAR-10), what about larger or more complex datasets like CIFAR-100. the task seem easy.
4.The paper lacks detailed computational efficiency, communication costs comparisons.
5.There is no ablation study to validate the effectiveness of the three key steps(local learning, graph inference, model update). The introduction of neural networks may deviate significantly from theoretical assumptions, and it remains unclear whether all three components are equally effective.

**Questions:**

1.Why are non-optimal results in Table 2 highlighted in bold?
2.How does DeLAMA perform on larger or more complex datasets, such as CIFAR-100 or ImageNet, compared to existing methods? Are there plans to evaluate its scalability and generalization capabilities in more challenging scenarios?
3.As mentioned in the weaknesses, could the authors conduct ablation studies to isolate the contributions of the three key components (local learning, graph inference, model update)? This would help clarify whether the introduction of neural networks significantly deviates from theoretical assumptions and whether all components are equally effective.
4. Has DeLAMA been compared with state-of-the-art methods developed after 2024?

---

### Official Review · Reviewer_wGXv · 2025-10-31

**Soundness:** 3
**Presentation:** 3
**Contribution:** 3
**Rating:** 6
**Confidence:** 2

**Summary:**

This work proposes a novel decentralized lifelong-adaptive collaborative learning framework based on numerical optimization and algorithm unrolling, named DeLAMA. It enables multiple agents to efficiently detect collaboration relationships and adapt to ever-changing observations.

**Strengths:**

The study appears to be theoretically sound and technically solid.

**Weaknesses:**

The limitation of this study is the simplicity of its experimental design. The evaluation is restricted to image classification and simulated robot mapping tasks. Assessing performance in real-world scenarios would provide a more comprehensive validation.

**Questions:**

See Weakness.

---

### Meta-Review · Area_Chair_WBSU · 2026-01-06

**Summary:**

Some of the limitations of this paper are summarized as follows:
1. Weak evaluations: insignificant performance improvement and old benchmarks
2. High computational complexity and latency for the optimization algorithm
3. Scalability issues

**Reviewer Concerns:**

No rebuttal

**Reviewer Scores:**

No change of scores

---

### Decision · Program_Chairs · 2026-01-26

Reject